# The kernel of graph indices for vector search

## Abstract

The most popular graph indices for vector search use principles from computational geometry to build the graph. Hence, their formal graph navigability guarantees are only valid in Euclidean space. In this work, we show that machine learning can be used to build graph indices for vector search in metric and non-metric vector spaces (e.g., for inner product similarity). From this novel perspective, we introduce the Support Vector Graph (SVG), a new type of graph index that leverages kernel methods to establish the graph connectivity and that comes with formal navigability guarantees valid in metric and non-metric vector spaces. In addition, we interpret the most popular graph indices, including HNSW and DiskANN, as particular specializations of SVG and show that new navigable indices can be derived from the principles behind this specialization. Finally, we propose SVG-L0 that incorporates an $\ell_0$ sparsity constraint into the SVG kernel method to build graphs with a bounded out-degree. This yields a principled way of implementing this practical requirement, in contrast to the traditional heuristic of simply truncating the out edges of each node. Additionally, we show that SVG-L0 has a self-tuning property that avoids the heuristic of using a set of candidates to find the out-edges of each node and that keeps its computational complexity in check.

## 1 Introduction

Vector search has become a critical component of AI infrastructure. For example, in retriever-augmented generation (RAG) (Lewis et al., 2020), vector search is used to ground knowledge and prevent hallucinations. The literature on vector search evolves quickly, trying to keep up with ever-increasing requirements: more vectors with larger dimensionality, higher speeds, and lower deployment costs, all while maintaining high accuracy. The most classical methods, trees (Beygelzimer et al., 2006; Navarro, 2002; Krauthgamer & Lee, 2004; Bentley, 1975) and hashing (Wang et al., 2018; Jafari et al., 2021), often struggle to achieve high accuracy at high speeds. Inverted indices (Muja & Lowe, 2014; Johnson et al., 2021) are simple to build, but involve a large number of similarity computations.

In recent years, graph-based indices (e.g., Dearholt et al., 1988; Arya & Mount, 1993; Malkov & Yashunin, 2020; Fu et al., 2019; Subramanya et al., 2019) strike an excellent balance of accuracy and speed and, as a consequence, have been widely deployed in the real world with great success. Here, a directed graph, where each vertex corresponds to a database vector and edges represent neighbor-relationships between vectors, is efficiently traversed to find the (approximate) nearest neighbors of a query vector in sublinear time. The graph edges need to be carefully selected to ensure that this traversal yields correct results (i.e., the graph is navigable). Starting with the seminal works by Dearholt et al. (1988) and Arya & Mount (1993), navigable graphs are built using computational geometry principles to perform edge selection. The Delaunay graph is a fully navigable triangulation (Wang et al., 2021) but it is too dense in higher dimensions. Informally, graph-building algorithms sparsify the graph by examining these triangles and only keeping a small subset. The Delaunay graph and the triangle pruning rules are defined in Euclidean space, which limits the navigability guarantees of the resulting graphs to this specific case. However, these algorithms are commonly used in non-Euclidean spaces, where their underlying principles do not hold, to build graphs that work well in practice but lack formal guarantees.

In this work, we analyze graph indices from a new perspective: we rely on machine learning instead of computational geometry. In particular, we study graph indices formally in light of kernel methods. In this setting, we have a similarity function $\text{sim}(\mathbf{x}, \mathbf{x}') : \mathbb{R}^d \times \mathbb{R}^d \to \mathbb{R}$ and an associated kernel $K(\mathbf{x}, \mathbf{x}') = h(\text{sim}(\mathbf{x}, \mathbf{x}'))$, where $h$ is a (possibly) nonlinear and monotonically non-decreasing function. Throughout this work, we assume that the kernel is positive semidefinite (PSD), i.e., the feature expansion $K(\mathbf{x}, \mathbf{x}') = \phi(\mathbf{x})^\top \phi(\mathbf{x}')$ is valid for (possibly infinite-dimensional) feature vectors $\phi(\mathbf{x})$. The exponential kernel is an important class of kernels, widely used in experimental and theoretical studies,

$$K_{\text{EXP}}(\mathbf{x}, \mathbf{x}') \stackrel{\text{def}}{=} \exp\left(\text{sim}(\mathbf{x}, \mathbf{x}')/\sigma^2\right), \tag{1}$$

where the hyperparameter $\sigma > 0$ is implicit. Two leading examples are given by the similarity functions

$$\text{sim}_{\text{EUC}}(\mathbf{x}, \mathbf{x}') \stackrel{\text{def}}{=} -\left\|\mathbf{x} - \mathbf{x}'\right\|_2^2, \tag{2}$$

$$\text{sim}_{\text{DP}}(\mathbf{x}, \mathbf{x}') \stackrel{\text{def}}{=} \mathbf{x}^\top \mathbf{x}' \tag{3}$$

that define the standard Radial Basis Function (a.k.a. Gaussian) and exponential dot product kernels, respectively. The similarity $\text{sim}_{\text{DP}}$ corresponds to the commonly used maximum inner product (MIP) retrieval problem. The exponential kernel is a natural choice, as it is used in the training loss (e.g., the entropy loss) of the embedding models that produce the vectors commonly used in practice (Radford et al., 2021; Karpukhin et al., 2020). Note that defining $\text{sim}(\mathbf{x}, \mathbf{x}') \stackrel{\text{def}}{=} -\text{dist}(\mathbf{x}, \mathbf{x}')^2$ for any distance function (e.g., Manhattan, Hamming, etc.) yields a valid exponential kernel.

Using kernels as our vantage point, we present the following contributions (all proofs in the appendix):

- We propose a new graph index, the Support Vector Graph (SVG). This new index arises from a novel perspective on graph indices (Section 2) and an accompanying formulation that models graph construction as a kernelized nonnegative least squares (NNLS) problem. This NNLS is equivalent to a support vector machine (SVM) whose support vectors provide the connectivity of the graph (Section 3).

- We provide formal results that show the navigability of the SVG for general PSD kernels (Section 3.1). Prior work established navigability for inner-product similarity via the inner-product Delaunay graph (Morozov & Babenko, 2018), which is infeasible to build in high dimensions. To our knowledge, SVG is the first concrete, bounded-degree construction that is provably navigable across general metric and non-metric similarities.

- We derive an interpretation of the most popular graph indices as SVG specializations, where the SVG optimization problem is used within the aforementioned traditional triangle pruning approach (Section 4). In particular, our results cover the popular HNSW (Malkov & Yashunin, 2020) and DiskANN (Subramanya et al., 2019). We also show that new triangle-pruning algorithms, valid in Euclidean and non-Euclidean spaces, can be derived from the principles behind this specialization. We prove that these algorithm variants lead to other new navigable indices.

- Finally, we address the construction of graphs with a bounded out-degree, a common feature in most practical deployments. For this, we propose SVG-L0 that includes a hard sparsity ($\ell_0$) constraint in the SVG optimization (Section 5). SVG-L0 yields a principled way of handling the requirement, in contrast with the traditional heuristic, which simply truncates the out edges of each node. Additionally, we show that SVG-L0 has a self-tuning property, which avoids setting a set of candidate edges for each graph node and yet still has a computational complexity sublinear in the number of indexed vectors.

This work centers on the formal analysis of SVG and SVG-L0. We complement the theory with empirical results that verify our navigability guarantees and characterize the proposed construction algorithms (Section 6). These experiments validate the theory; large-scale system benchmarking is a distinct engineering effort that we leave to future work. For reproducibility, we make our implementation available at `[anonymized_url]`.

**Notation.** We denote the set of natural numbers from 1 to $n$ by $[1 \ldots n]$. We denote vectors/matrices by lowercase/uppercase bold letters, e.g., $\mathbf{v} \in \mathbb{R}^n$ and $\mathbf{A} \in \mathbb{R}^{m \times n}$. Individual entries of a matrix $\mathbf{A}$ (resp. vector $\mathbf{v}$) are denoted by $\mathbf{A}_{[ij]}$ (resp. $v_i$). The $i$-th row and column of $\mathbf{A}$ are denoted by $\mathbf{A}_{[i:]}$ and $\mathbf{A}_{[:i]}$, respectively.

---

**Algorithm 1:** Greedy graph search

---

**Input** : Query $\mathbf{q} \in \mathbb{R}^d$, dataset $\left\{\mathbf{x}_i \in \mathbb{R}^d\right\}_{i=1}^n$, graph $G = ([1 \dots n], \mathcal{E})$, entry point $i_{\mathrm{ep}}$.
**Output:** Approximate nearest neighbor $i^*$.

**1** $i^* \leftarrow i_{\mathrm{ep}}$;
**2 Repeat**
**3** $\quad$ $i \leftarrow \underset{j \in \mathcal{N}_{i^*}}{\mathrm{argmax}} \, \mathrm{sim}(\mathbf{q}, \mathbf{x}_j)$; // $\mathcal{N}_{i^*}$ is the neighborhood of $i^*$
**4** $\quad$ **if** $\mathrm{sim}(\mathbf{q}, \mathbf{x}_i) > \mathrm{sim}(\mathbf{q}, \mathbf{x}_{i^*})$ **then** $i^* \leftarrow i$; // progress, continue
**5** $\quad$ **else return** $i^*$; // no progress, exit

---

The matrix containing a subset $\mathcal{I} \subset [1 \dots m]$ (resp. $\mathcal{J} \subset [1 \dots n]$) of the rows (resp. columns) of $\mathbf{A} \in \mathbb{R}^{m \times n}$ is denoted by $\mathbf{A}_{[\mathcal{I}:]}$ (resp. $\mathbf{A}_{[:\mathcal{J}]}$). A directed graph $G = ([1 \dots n], \mathcal{E})$ is composed by the node set $[1 \dots n]$ and the edge/vertex set $\mathcal{E}$, i.e., a set of ordered pairs $\vec{ij}$ with $i, j \in [1 \dots n]$. We define the neighborhood of node $i$ as $\mathcal{N}_i \overset{\text{def}}{=} \left\{ j \mid \vec{ij} \in \mathcal{E} \right\}$. A path $[v_1, \cdots, v_l]$ in $G$ is a list of nodes such that $(\forall i = 1, \cdots, l-1)$ $\overrightarrow{v_i v_{i+1}} \in \mathcal{E}$.

## 2 Graph indices for vector search in Euclidean Space

Using navigable graphs for vector search has a long history (Dearholt et al., 1988; Arya & Mount, 1993) but only became prominent in the last ten years (e.g., Subramanya et al., 2019; Malkov & Yashunin, 2020) with the increasing scale of unstructured data. Navigability is defined as the ability to reach any node when conducting a greedy graph traversal (Algorithm 1) using that node as the query. The following definitions from the literature formalize this concept for the Euclidean distance. In this case, the similarities in Algorithm 1 are transformed into Euclidean distances by switching the maximization of the similarity with the minimization of the distance.

**Definition 1** (Monotonic Path (Fu et al., 2019)). *Given a set of $n$ vectors $\left\{\mathbf{x}_i \in \mathbb{R}^d\right\}_{i=1}^n$, let $G = ([1 \dots n], \mathcal{E})$ denote a directed graph and $s, t \in [1 \dots n]$ be two nodes of $G$. A path $[v_1, \cdots, v_l]$ from $s = v_1$ to $t = v_l$ in $G$ is a monotonic path if and only if $(\forall i = 1, \cdots, l-1)$ $\|\mathbf{x}_{v_i} - \mathbf{x}_t\|_2 > \|\mathbf{x}_{v_{i+1}} - \mathbf{x}_t\|_2$.*

**Definition 2** (Monotonic Search Network (Fu et al., 2019)). *Given a set of $n$ vectors $\left\{\mathbf{x}_i \in \mathbb{R}^d\right\}_{i=1}^n$, a graph $G = ([1 \dots n], \mathcal{E})$ is a monotonic search network if and only if there exists at least one monotonic path from $s$ to $t$ for any two nodes $s, t \in [1 \dots n]$.*

A Monotonic Search Network (MSNet) is a navigable graph as demonstrated by the following lemma.

**Lemma 1** (Fu et al., 2019). *Let $G = ([1 \dots n], \mathcal{E})$ be a monotonic search network. Let $s, t \in [1 \dots n]$, then Algorithm 1 with $\mathbf{x}_t$ as the query and $s$ as the entry point finds a monotonic path from $s$ to $t$ in $G$.*

The Delaunay graph (DG) is an MSNet (Kurup, 1992). For a set $\mathcal{P}$ of points in Euclidean space, the DG is a triangulation such that no point in P is inside the circum-hypersphere of any of its triangles. For a graph with $n$ nodes, the number of edges in the DG rapidly approaches $O(n^2)$ as the dimensionality grows, limiting its usability for large datasets with high-dimensional vectors (the memory and computational complexities approach $O(n^2)$ and $O(n^{\lceil d/2 \rceil})$ (McMullen, 1970), respectively). As a consequence, many graph construction algorithms (e.g., Dearholt et al., 1988; Arya & Mount, 1993; Malkov & Yashunin, 2020; Fu et al., 2019; 2022; Subramanya et al., 2019) have been proposed over the years, operating under the (sometimes implicit) principle of sparsifying the DG. These algorithms work as depicted in Algorithm 2. For each node $i$, a candidate pool is selected (this is commonly implemented as an approximate nearest neighbor search), and then a pruning algorithm is used to select a maximally diverse set of nodes (i.e., far away from each other) while being close to $i$, see Figure 1. In essence, these algorithms were carefully designed to analyze the DG triangles (or a superset (Subramanya et al., 2019)) and discard those edges that are redundant for navigability. While these graphs were designed to have formal navigability guarantees when the candidate set $\mathcal{C}_i = [1 \dots n] \setminus \{i\}$ in Algorithm 2, they are lost when $\mathcal{C}_i \subset [1 \dots n] \setminus \{i\}$.

The DG and the main graph indices (e.g., Malkov & Yashunin, 2020; Fu et al., 2019; Subramanya et al., 2019) rely on principles from computational geometry, such as triangular inequalities and (as we show in

**Algorithm 2:** Graph index construction

    **Input**   : Dataset $\left\{\mathbf{x}_i \in \mathbb{R}^d\right\}_{i=1}^n$.
    **Output:** Graph $G = ([1 \ldots n], \mathcal{E})$.

**1** $\mathcal{E} \leftarrow \emptyset$;

**2 for** $i \in [1 \ldots n]$ **do**

**3**     **Selection:** choose a candidate pool $\mathcal{C}_i \subseteq [1 \ldots n] \setminus \{i\}$;

**4**     **Pruning:** create set $\mathcal{N}_i \subseteq \mathcal{C}_i$ containing the out neighbors of node $i$ by applying a pruning algorithm;

**5**     $\mathcal{E} \leftarrow \mathcal{E} \cup \left\{\vec{ij} \,\middle|\, j \in \mathcal{N}_i\right\}$;

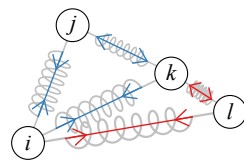

Figure 1: Conceptual depiction of the pruning strategy in Euclidean space to find the out-edges of node $i$ in the graph index $G = ([1 \ldots n], \mathcal{E})$. Attractive (inward arrowheads) and repulsive (outward arrowheads) forces promote similarity with $i$ or diversity between candidates, respectively. Blue and red arrows depict favorable and less favorable forces, respectively. Here, $\{\vec{ij}, \vec{ik}\} \subset \mathcal{E}$ but $\vec{il} \notin \mathcal{E}$ as one can move from $i$ to $k$ and then from $k$ to $l$ using the greedy search in Algorithm 1.

Section 4) on the law of cosines, which are only valid in Euclidean space. These graph indices have been used in non-Euclidean vector spaces by extending their edge pruning rules to other similarities in an ad hoc fashion, resulting in a lack of understanding of their practical behavior. For example, ip-NSW (Morozov & Babenko, 2018) extends the NSW construction to inner-product similarity by connecting each node to its highest-inner-product neighbors; while effective in practice, this construction is acknowledged to lack a formal navigability guarantee.

### 2.1 From graph search to multiclass classification

We now adopt an alternative viewpoint that will ultimately lead to new developments. For this, we think of a greedy search using Algorithm 1 in DG, the original monotonic search network, as a multiclass classification problem, as explained next.

The Voronoi diagram is a tessellation of the space, where each node $i$ of the DG corresponds to a distinct convex cell $C_i$ (see Figure 16 in the appendix). Two nodes are connected in the DG if the corresponding Voronoi cells share a facet. We associate with each cell $C_i$ a decision function $f_i : \mathbb{R}^d \to \mathbb{R}$ such that: $f_i(\mathbf{x}) \geq 0$ if $\mathbf{x} \in C_i$, $f_i(\mathbf{x}) < 0$ if $\mathbf{x} \notin C_i$, and $f_i(\mathbf{x})$ decreases as the distance between $\mathbf{x}$ and $C_i$, $\min_{\mathbf{x}' \in C_i} \|\mathbf{x} - \mathbf{x}'\|_2$, increases. Each $f_i$ is determined by the intersection of the half-spaces of the boundaries of $C_i$. Finding the nearest neighbor of a query $\mathbf{q}$ is equivalent to finding $i$ such that $f_i(\mathbf{q}) \geq 0$. This corresponds to a multiclass classification problem with $n = |\mathcal{X}|$.[1] Of course, this is not computationally very useful, as we are evaluating $n$ classifiers (i.e., scanning the entire set $\mathcal{X}$). Seeking fewer evaluations, we conceptualize Algorithm 1 as follows: if $f_{i^*}(\mathbf{q}) \geq 0$, the vector $\mathbf{x}_{i^*}$ is the nearest neighbor of $\mathbf{q}$; if $f_{i^*}(\mathbf{q}) < 0$, move to the adjacent cell $i$ so that $f_i(\mathbf{q})$ is maximum. That is, instead of directly solving the multiclass classification problem, we use the decision functions of adjacent cells (i.e., given by the Delaunay edges) to find an ascending path $[v_1, \cdots, v_l]$ such that $f_{v_i} < f_{v_{i+1}}$. Through this ascent algorithm, we only evaluate a small subset of the $n$ classifiers.

This qualitative viewpoint raises several questions. Can we use machine learning (ML) to build graph indices? And in non-Euclidean spaces? Can we leverage ML to build parsimonious graphs? Is there a connection between the ML approach and "traditional" graph indices? In the remainder of this paper, we answer these questions in the affirmative using support vector machines to develop new graph indices in Euclidean and non-Euclidean spaces with the properties and contributions discussed in the introduction.

## 3 The Support Vector Graph

We now define a new type of graph inspired by the ideas in Section 2. Instead of relying on principles from computational geometry, we directly leverage the result of an optimization algorithm, the nonnegative least squares problem in kernel space. The positive semidefinite kernel matrix $\mathbf{K}$ with entries $\mathbf{K}_{[ij]} = K(\mathbf{x}_i, \mathbf{x}_j) =$

---

[1]Inverted indices use a similar computational motif derived from Voronoi diagrams (Jégou et al., 2011).

$\phi(\mathbf{x}_i)^\top \phi(\mathbf{x}_j)$ can be written as $\mathbf{K} = \boldsymbol{\Phi}^\top \boldsymbol{\Phi}$ where $\boldsymbol{\Phi} = [\phi(\mathbf{x}_1), \cdots, \phi(\mathbf{x}_n)]$, with the vectors horizontally stacked. From now on, and unless otherwise specified, the similarity between two vectors $\mathbf{x}_i$ and $\mathbf{x}_j$ will only be determined by the value of $K(\mathbf{x}_i, \mathbf{x}_j)$. With these elements, we present the proposed graph index.

**Definition 3.** *We define the Support Vector Graph (SVG) as the result of connecting node $i$ to the non-zero elements of the minimizer $\mathbf{s}^{(i)}$ of*

$$\min_{\mathbf{s}} \frac{1}{2} \|\phi(\mathbf{x}_i) - \boldsymbol{\Phi}\mathbf{s}\|_2^2 \quad s.t. \quad \mathbf{s} \geq \mathbf{0}, \, s_i = 0, \|\mathbf{s}\|_2^2 \leq n^{-1}, \tag{4}$$

*where $\boldsymbol{\Phi} = [\phi(\mathbf{x}_1), \cdots, \phi(\mathbf{x}_n)]$ are the stacked feature vectors of a PSD kernel $K$.*

In essence, Problem (4) can be seen as finding the projection of $\phi(\mathbf{x}_i)$ onto a convex set,

$$\operatorname*{argmin}_{\mathbf{y}} \|\phi(\mathbf{x}_i) - \mathbf{y}\|_2^2 \quad \text{s.t.} \quad \mathbf{y} \in \mathrm{sc}(\mathcal{X}, i), \tag{5}$$

where

$$\mathrm{sc}(\mathcal{X}, i) \stackrel{\text{def}}{=} \left\{ \sum_{j=1}^n s_j \phi(\mathbf{x}_j) \,|\, s_j \geq 0, s_i = 0, \sum_{j=1}^n s_j^2 \leq n^{-1} \right\}. \tag{6}$$

The set $\mathrm{sc}(\mathcal{X}, i)$ is the conical combination of the vectors in $\mathcal{X} \setminus \{\phi(\mathbf{x}_i)\}$ with an additional constraint on the squares of the weights $s_j$. This set is convex, making Problem (4) convex. A distinctive feature of Problem (4), relative to other graph constructions based on nonnegative least squares (e.g., Shekkizhar & Ortega, 2023), is the weight-budget constraint $\|\mathbf{s}\|_2^2 \leq n^{-1}$. This is a substantive modeling difference rather than a technicality: it is precisely what yields the navigability guarantee of the next section, where it ensures $\mathbf{1}^\top \mathbf{s}^* \leq 1$ (by Cauchy–Schwarz over the at most $n$ nonzero entries of $\mathbf{s}^*$), a property on which the navigability proof relies. The bound must moreover be imposed by a *super-linear* constraint: Lemma 3 shows that any $\ell_p$ budget with $p > 1$ preserves the navigability argument, whereas a linear ($\ell_1$) constraint $\mathbf{1}^\top \mathbf{s} \leq c$ does not; we use the $\ell_2$ form as the simplest admissible choice. When the budget is active, which is often the case since $n^{-1}$ is small, it genuinely shapes the minimizer, altering both its support and its weights relative to the unconstrained conical projection rather than merely rescaling it.

When $K(\mathbf{x}_i, \mathbf{x}_j) = \phi(\mathbf{x}_i)^\top \phi(\mathbf{x}_j)' \geq 0$ for any $i, j$ (a common choice), the angle between any pair of vectors is less than $\pi/2$ and we can find a rotation such that all the feature vectors lie in the positive orthant. Then, the columns of $\boldsymbol{\Phi}$ satisfy the conditions that ensure a unique solution (e.g., Wang & Tang, 2009; Slawski & Hein, 2011; 2014). Moreover, in this setting, Problem (4) is self-regularizing, in the sense that its minimizer is naturally sparse (Slawski & Hein, 2011) without including any explicit constraints promoting sparsity (in contrast to the DG whose sparsity depends on the input dimensionality, i.e., less sparse at higher dimensions).

The use of sparsity-regularized regression problems to build graphs is not new in machine learning. Some notable applications include the estimation of sparse inverse covariance matrices (Meinshausen & Bühlmann, 2006), subspace learning and clustering (Cheng et al., 2010; Hosseini & Hammer, 2018), spectral clustering (Xiao et al., 2012), and nonnegative matrix factorization (for bipartite graphs) (Kumar et al., 2013). In particular, a variant of Problem (4), which relies on the selection of a candidate pool as in Algorithm 2, was used for manifold learning (Shekkizhar & Ortega, 2023). However, our analysis of graphs built with nonnegative sparse regression for vector search is new.

None of the aforementioned methods target the construction of graph *indices for vector search* and therefore do not carry navigability guarantees, which are the central objects of interest in this paper. Beyond this difference in application, our contribution is conceptually broader than the sparse-regression formulation itself. First, Problem (4) admits a support-vector interpretation (Theorem 1): the support vectors of an equivalent SVM classifier define each node's out-edges, a connection absent from the sparse/kernel graph-construction methods above. Second, we establish formal navigability guarantees for the resulting graph that extend beyond Euclidean distance to general metric and non-metric similarities (Section 3.1), a guarantee neither NNK nor the other constructions above provide. Third, this same formulation recovers HNSW, MRNG, and Vamana as special cases (Section 4), unifying constructions previously understood only through

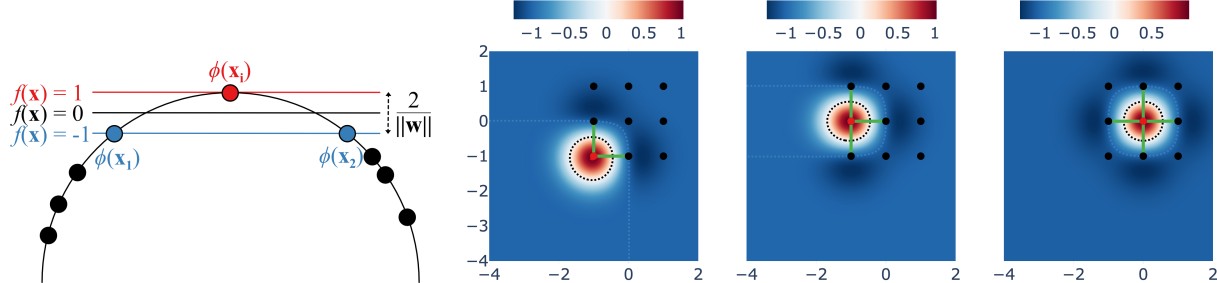

Figure 2: (Left) Conceptual representation of the SVM hyperplane and margins involved in SVG. Here, the vector $\mathbf{x}_i$ is connected to its support vectors $\mathbf{x}_1$ and $\mathbf{x}_2$ for which $f_i(\mathbf{x}_1) = f_i(\mathbf{x}_2) = -1$, see Equation (9). (Right) Example of the SVM decision function values (the level sets $f_i(\mathbf{x}) = 1, 0, -1$ are marked in dotted red, black and blue lines, respectively). We observe that the function $f_i$ adjusts its shape to the topology of its surrounding points (i.e., the area where $f_i(\mathbf{x}) > 0$ adapts to its surroundings).

separate, ad hoc pruning heuristics. Finally, SVG-L0 (Section 5) turns the formulation into a bounded-out-degree, candidate-pool-free construction algorithm, addressing the practical limitations, i.e., candidate-pool selection and unconstrained sparsity, that NNK and the other constructions above share.

Furthermore, the SVG establishes an interesting link between graph indices and parsimonious vector coding. That is, with a linear kernel, the loss in Problem (4) becomes $\frac{1}{2} \|\mathbf{x}_i - \mathbf{X}\mathbf{s}\|_2^2$, where $\mathbf{X} = [\mathbf{x}_1, \cdots, \mathbf{x}_n]$. This formulation is commonly used to represent (e.g., Elhamifar et al., 2012) and quantize vectors (in additive quantization (Martinez et al., 2016), for example). There is a conceptual parallelism with inverted indices (Jégou et al., 2011), which are derived from vector quantizers (k-means).

As we show next, the connection between Problem (4) and navigable graphs starts emerging as we dig deeper into the problem's properties. By analyzing the expanded form of Problem (4),

$$\min_{\{s_j\}_{j=1}^n} \frac{1}{2} K(\mathbf{x}_i, \mathbf{x}_i) + \underbrace{\frac{1}{2} \sum_{j,k \neq i} s_j s_k K(\mathbf{x}_j, \mathbf{x}_k)}_{\text{term A}} - \underbrace{\sum_{j \neq i} s_j K(\mathbf{x}_i, \mathbf{x}_j)}_{\text{term B}} \quad \text{s.t.} \quad (\forall j)\, s_j \geq 0,\ s_i = 0, \|\mathbf{s}\|_2^2 \leq n^{-1}, \quad (7)$$

it becomes clear that its solution balances diversity (repulsion) and similarity (attraction) forces using similar principles as those shown in Figure 1 and analyzed in detail in Section 4 for other popular graph indices. The minimization of term A promotes the selection of a diverse set of edges, i.e., indices $j, k$ such that $K(\mathbf{x}_j, \mathbf{x}_k)$ is small. When using the RBF kernel, it favors out-neighbors that are far away from each other. The minimization of term B promotes the selection of edges that are similar to $\mathbf{x}_i$, i.e., indices $j$ such that $K(\mathbf{x}_i, \mathbf{x}_j)$ is large. When using the RBF kernel, it favors out-neighbors that are close to $\mathbf{x}_i$.

In the previous section, we qualitatively connected graph indices with a multi-class classification problem. It turns out that Problem (4) is a classification problem disguised as a regression problem.

**Theorem 1.** *Problem (4) is equivalent to a hard-margin support vector machine classifier using the labels*

$$y_j = \begin{cases} 1 & \text{if } j = i, \\ -1 & \text{otherwise.} \end{cases} \quad (8)$$

*The nonzero elements of the minimizer $\mathbf{s}^{(i)}$ of Problem (4) are the support vectors.*

We refer the reader to Appendix A for a quick primer on SVMs. Let $\mathbf{s}^*$ be the minimizer of Problem (4) for $\mathbf{x}_i$. Let $\phi(\mathbf{x}_{\text{SV}})$ be any of the support vectors of our SVM problem (corresponding to a $j'$ such that $s_{j'}^* > 0$).

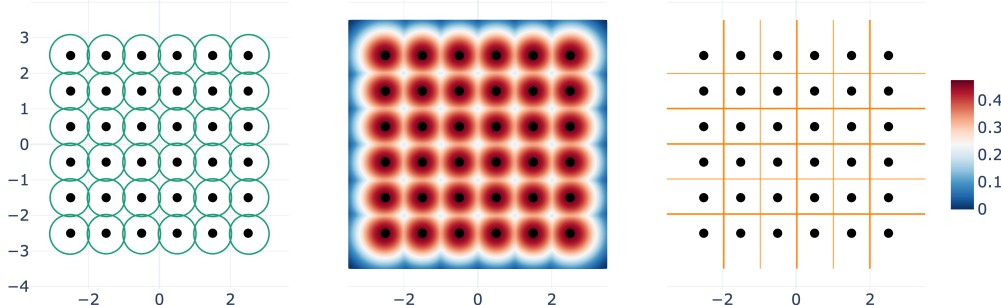

Figure 3: The SVM decision boundaries (left), i.e., $f_i(\mathbf{x}) = 0$, for each point in a regular 2D grid; see Equation (9). The function $f(\mathbf{x}) = \max_i f_i(\mathbf{x})$ (center) induces a tessellation. As expected, the tessellation, found by running a watershed algorithm on $f(\mathbf{x})$, forms a regular grid (right).

The SVM decision function is defined as

$$f_i(\mathbf{x}) = \frac{1}{c} \left( \mathbf{w}_i^\top \phi(\mathbf{x}) + b_i \right) \quad \text{where} \quad \begin{aligned} \mathbf{w}_i &= \phi(\mathbf{x}_i) - \sum_{j \neq i} s_j^* \phi(\mathbf{x}_j), \\ b_i &= -\frac{1}{2} \mathbf{w}_i^\top \left( \phi(\mathbf{x}_i) + \phi(\mathbf{x}_{\text{SV}}) \right), \\ c &= \frac{1}{2} \mathbf{w}_i^\top \left( \phi(\mathbf{x}_i) - \phi(\mathbf{x}_{\text{SV}}) \right). \end{aligned} \tag{9}$$

The full derivation can be found in Appendix B. Here, $b_i$ is the SVM bias that centers the decision function so that its margins pass through $\mathbf{x}_i$ (where $f_i = 1$) and through the support vectors (where $f_i = -1$). By construction, $f_i(\mathbf{x}_i) = 1$ and $f_i(\mathbf{x}_j) = -1$ for every $j \in \mathcal{N}_i$, and $f_i(\mathbf{x}_j) < -1$ for every $j \in [1..n] \setminus (\mathcal{N}_i \cup \{i\})$. In Figure 2 (left), we present a conceptual representation of these level sets as hyperplanes in feature space. Figure 2 (right) illustrates that these level sets materialize in the original space as nonlinear boundaries that adapt their shape to the topology of the vectors surrounding $\mathbf{x}_i$.

This alternative formulation of Problem (4) makes it easy to see why $|\mathcal{N}_i| \ll n$: The support vector set $\mathcal{N}_i$ is sparse in separable and non-degenerate settings. Although we leave formal sparsity results for future work, we illustrate this point with an important example. For $D$-dimensional feature vectors, the number of support vectors in SVMs is at most $D + 1$. In MIP (maximum inner product) retrieval, possibly the most common vector search today, the kernel is linear $K(\mathbf{x}, \mathbf{y}) = \mathbf{x}^\top \mathbf{y}$. In this case, each node in the SVG will have $d + 1$ out-edges at most for $\mathbf{x} \in \mathbb{R}^d$. The dimensionalities used today ($d = 1024$ to $d = 4096$) imply a sparse graph for $n \gg d$. This $D + 1$ bound applies to any kernel with a finite-dimensional feature map (e.g., linear or polynomial kernels). For kernels with infinite-dimensional features, such as the exponential/RBF kernel used for Euclidean similarity,[2] no such a-priori bound exists; there, sparsity follows instead from the separability and non-degeneracy of the data, and the out-degree grows with dimension (Figure 5).

The decision functions $f_i$ induce a tessellation of the space, as observed in Figure 3. We can find this tessellation by considering the function $F(\mathbf{x}) = -\max_i f_i(\mathbf{x})$ as a topographic map and separating adjacent catchment basins (following its gradient) using a watershed algorithm (Couprie & Bertrand, 1997). The link between the SVG and this tessellation is analogous to that of the Delaunay graph and the Voronoi diagram.

SVG also shares a deep connection with the DG. Other graph indices are subgraphs of the DG by applying pruning rules to its edges. As shown next, when using a kernel based on the Euclidean distance (e.g., the RBF kernel), SVG sparsifies the DG by solving optimization problems (see Figures 4 and 5).

**Theorem 2.** *Let $G$ be the Delaunay graph computed from the original vectors $\{\mathbf{x}_i\}_{i=1}^n$. When using a kernel based on the Euclidean distance (e.g., RBF), the support of the solution to Problem (4) is a subset of the neighbors of node $i$ in $G$.*

---

[2]The exponential kernel has an infinite series expansion and thus infinite-dimensional features. It is only one possible choice: selecting a kernel with a finite-dimensional feature map, such as a polynomial kernel, keeps the $D + 1$ out-degree bound in force.

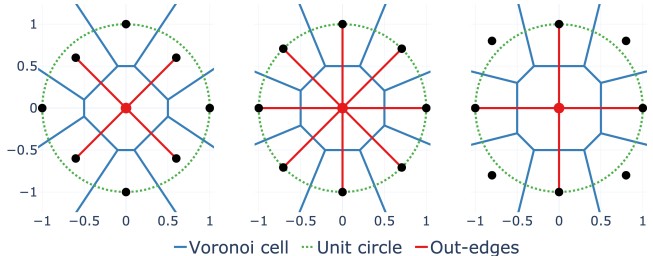

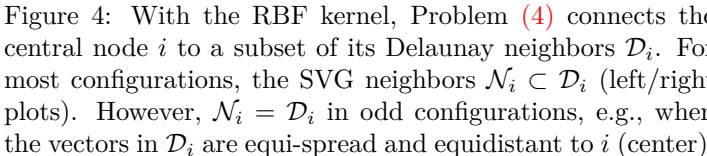

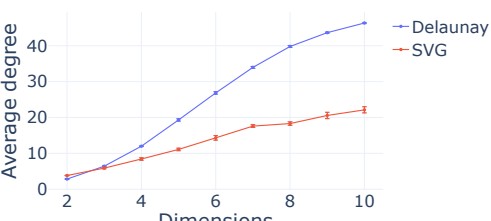

Figure 4: With the RBF kernel, Problem (4) connects the central node $i$ to a subset of its Delaunay neighbors $\mathcal{D}_i$. For most configurations, the SVG neighbors $\mathcal{N}_i \subset \mathcal{D}_i$ (left/right plots). However, $\mathcal{N}_i = \mathcal{D}_i$ in odd configurations, e.g., when the vectors in $\mathcal{D}_i$ are equi-spread and equidistant to $i$ (center).

Figure 5: The average cardinality of the SVG neighbors grows slower than that of the Delaunay neighbors for (ten realizations of) 100 randomly distributed vectors as their dimension grows.

To conclude this section, we briefly analyze the setting where we have an indefinite kernel. These kernels do not induce a Reproducing Kernel Hilbert Space (RKHS) and consequently do not have features $\phi(\mathbf{x})$. We could still derive a graph from such kernels by starting from Problem (7). Multiple differences arise in this scenario. First, the interpretation of Problem (7) as a NNLS and/or a SVM classifier are lost. Much of the geometry insights that we gain from such perspectives are foregone too. Second, Problem (7) becomes non-convex and there might be multiple edge configurations that are valid solutions. Lastly, but perhaps more interestingly, the graph remains navigable as the results in the next section are still valid in non-RKHS settings.

## 3.1 Navigability

So far, we have described how to build an SVG and how it shares some key properties with the DG and other graph indices. We now turn our attention to the analysis of its navigability, showing that SVGs are navigable for general kernels (all proofs are in the appendix).

There is abundant literature (e.g., Dearholt et al., 1988; Arya & Mount, 1993; Malkov & Yashunin, 2020; Fu et al., 2019; 2022; Subramanya et al., 2019) on graph indices that were designed to have formal navigability guarantees only in the Euclidean case, with this guarantee forgone when using other similarities. Before proceeding, we provide new definitions that generalize the notion of graph navigation for arbitrary similarities. With non-metric similarities, the terminal node may be different from the query. That is, whereas $\text{sim}_{\text{EUC}}(\mathbf{x}_j, \mathbf{x}_j) = \max\limits_{i=1,\dots,n} \text{sim}_{\text{EUC}}(\mathbf{x}_i, \mathbf{x}_j)$ for $j \in [1 \dots n]$ is always true, $\text{sim}_{\text{DP}}(\mathbf{x}_j, \mathbf{x}_j) < \max\limits_{i=1,\dots,n} \text{sim}_{\text{DP}}(\mathbf{x}_i, \mathbf{x}_j)$ is possible if $\exists i \in [1 \dots n]$ such that $\mathbf{x}_i = c\mathbf{x}_j$ for $c > 1$ (see Equations (2) and (3)).

**Definition 4** (Generalized Monotonic Path). *Given a set of $n$ vectors $\left\{\mathbf{x}_i \in \mathbb{R}^d\right\}_{i=1}^n$ and the similarity function* sim, *let $G = ([1\dots n], \mathcal{E})$ denote a directed graph, $s, k \in [1 \dots n]$ be two nodes of $G$, and $t = \text{argmax}\limits_{i=1,\dots,n} \text{sim}(\mathbf{x}_i, \mathbf{x}_k)$. A path $[v_1, \cdots, v_l]$ from $s = v_1$ to $t = v_l$ in $G$ is a generalized monotonic path if and only if $(\forall i = 1, \cdots, l-1) \, \text{sim}(\mathbf{x}_{v_i}, \mathbf{x}_t) < \text{sim}(\mathbf{x}_{v_{i+1}}, \mathbf{x}_t)$.*

With this change, the generalization of a navigable graph follows naturally.

**Definition 5** (Generalized Monotonic Search Network). *Given a set of $n$ vectors $\left\{\mathbf{x}_i \in \mathbb{R}^d\right\}_{i=1}^n$ and the similarity function* sim, *a graph $G = ([1 \dots n], \mathcal{E})$ is a generalized monotonic search network if and only if there exists at least one generalized monotonic path from $s$ to $t = \text{argmax}\limits_{i=1,\cdots,n} \text{sim}(\mathbf{x}_i, \mathbf{x}_k)$ for any two nodes $s, k \in [1 \dots n]$.*

Definition 5 is equivalent to Definition 2 when using a similarity based on the Euclidean distance as $t = k$.

A generalized monotonic path establishes that a route exists between any two nodes; it does not, by itself, say anything about whether a search *algorithm* finds that route. We separate these two questions cleanly. We first isolate the graph-theoretic property that makes such a path constructible, stated for an arbitrary

reference node rather than tied to any specific query, so that it can be verified once for the SVG and reused for any downstream algorithmic argument.

**Definition 6** (No-close-disconnected-node property). *A directed graph $G = ([1 \ldots n], \mathcal{E})$ has the* no-close-disconnected-node *property at reference $r \in [1 \ldots n]$ if for every $i \neq r$, $r \notin \mathcal{N}_i$ implies $\exists j \in \mathcal{N}_i$ with $K(\mathbf{x}_j, \mathbf{x}_r) > K(\mathbf{x}_i, \mathbf{x}_r)$.*

As a partial step towards a formal navigability result, using the optimality conditions of Problem (4), we first show in Lemma 2 that the SVG has the no-close-disconnected-node property at every reference node, i.e., (i) there is always an edge in the SVG that improves the current similarity toward that reference, or (ii) we have reached our objective.

**Lemma 2.** *The SVG has the no-close-disconnected-node property (Definition 6) at every reference $r \in [1 \ldots n]$: for $i \neq r$, $\exists j \in \mathcal{N}_i$ such that $K(\mathbf{x}_i, \mathbf{x}_r) < K(\mathbf{x}_j, \mathbf{x}_r)$.*

The role of the budget constraint $\|\mathbf{s}\|_2^2 \leq n^{-1}$ in the proof of Lemma 2 is precise, and it explains why an $\ell_2$ (rather than $\ell_1$) constraint is used.

**Lemma 3** (The budget constraint must be super-linear). *Suppose the budget constraint $\|\mathbf{s}\|_2^2 \leq n^{-1}$ in Problem (4) is replaced by $\rho(\mathbf{s}) \leq c$ for a convex, differentiable $\rho : \mathbb{R}_{\geq 0}^n \to \mathbb{R}$ and some $c > 0$. The proof of Lemma 2 uses only two properties of the minimizer $\mathbf{s}^*$:*

*(i) $\mathbf{1}^\top \mathbf{s}^* \leq 1$; and*
*(ii) $\sum_j s_j^* K(\mathbf{x}_j, \mathbf{x}_r) \geq K(\mathbf{x}_i, \mathbf{x}_r)$ for every $r \notin \mathcal{N}_i$.*

*Property (ii) holds whenever $\partial \rho / \partial s_r |_{s_r = 0} = 0$ for all $r$. For $\rho(\mathbf{s}) = \|\mathbf{s}\|_p^p$ this condition holds for every $p > 1$ and fails for $p = 1$; the choice $p = 2$ with $c = n^{-1}$ moreover guarantees (i). Consequently a linear ($\ell_1$) budget $\mathbf{1}^\top \mathbf{s} \leq c$ does* not *preserve the navigability guarantee, whereas any $\ell_p$ budget with $p > 1$ (in particular the $\ell_2$ budget used throughout) does.*

Analyzing the inequality in Lemma 2 in terms of the feature vectors can help understand its geometric meaning: $K(\mathbf{x}_i, \mathbf{x}_r) < K(\mathbf{x}_j, \mathbf{x}_r)$ implies that $\phi(\mathbf{x}_r)^\top (\phi(\mathbf{x}_j) - \phi(\mathbf{x}_i)) > 0$, where the vector $\phi(\mathbf{x}_j) - \phi(\mathbf{x}_i)$ can be interpreted as a "discrete gradient" that needs to be positively correlated with the reference vector $\phi(\mathbf{x}_r)$.

Equipped with these new concepts, we provide our main theoretical result, the navigability of the SVG.

**Theorem 3.** *An SVG is a generalized monotonic search network (Definition 5).*

This follows directly from Lemma 2: instantiated at reference $r = t := \operatorname{argmax}_i K(\mathbf{x}_i, \mathbf{x}_k)$ for any query $k$, it shows that from any node $i \neq t$ not yet connected to $t$, some neighbor strictly improves similarity to $\mathbf{x}_t$; repeating this (the graph is finite) produces a generalized monotonic path. Theorem 3 is therefore a statement purely about the existence of such a path in the graph SVG builds. It does not imply that a search algorithm, which does not know $t$ in advance, is guaranteed to find it. We turn to that question next.

The SVG, derived using completely different tools (i.e., kernel methods from machine learning) than existing graph indices, is a suitable graph index for vector search. The closest prior result, ip-NSW (Morozov & Babenko, 2018), shows that the inner-product Delaunay graph is navigable, generalizing the classical Euclidean Delaunay result to a single non-metric similarity. However, that graph is infeasible to construct in high dimensions, and the practical ip-NSW approximation (connecting each node to its highest-inner-product neighbors) forgoes the guarantee.

### 3.1.1 From path existence to a search guarantee

Algorithm 1 does not know $t$; it only ever has access to the query $\mathbf{x}_k$, and it greedily maximizes similarity *to the query*, not to the (unknown) target (in contrast with Definition 4 that implies knowledge of the target). In the classical Euclidean case this distinction is invisible because a point is always its own nearest neighbor, so $t = k$ and the two notions of progress coincide. For a general similarity, $t$ can differ from $k$, and the two notions of progress, toward $\mathbf{x}_t$ (what Lemma 2 controls) and toward $\mathbf{x}_k$ (what Algorithm 1 actually optimizes), need not agree. We now characterize what Algorithm 1 is actually guaranteed to reach.

**Definition 7** (Dominance set). *For a query index $k$, the* dominance set *is*

$$\mathcal{D}_k := \{i \in [1 \ldots n] \mid K(\mathbf{x}_i, \mathbf{x}_k) \geq K(\mathbf{x}_k, \mathbf{x}_k)\} .$$

*Note $k \in \mathcal{D}_k$ trivially and $t := \operatorname{argmax}_i K(\mathbf{x}_i, \mathbf{x}_k) \in \mathcal{D}_k$ by definition of $t$, so $\mathcal{D}_k \neq \emptyset$; moreover $\mathcal{D}_k = \{k\}$ if and only if $t = k$.*

**Lemma 4** (Query-driven convergence). *Let $G$ be any directed graph with the no-close-disconnected-node property (Definition 6) at reference $k$. Then for any entry point $s$, Algorithm 1 run on $G$ with $\mathbf{x}_k$ as the query and $s$ as the entry point terminates at some node in $\mathcal{D}_k$.*

By Lemma 2, the SVG has the no-close-disconnected-node property at every reference, in particular at $k$; Lemma 4 therefore applies to it directly.

**Corollary 1** (Exact recovery for self-dominant similarities). *If $\mathbf{x}_k$ is its own best match, i.e. $K(\mathbf{x}_k, \mathbf{x}_k) \geq K(\mathbf{x}_i, \mathbf{x}_k)$ for all $i$ (equivalently $\mathcal{D}_k = \{k\}$, equivalently $t = k$), then Algorithm 1 on the SVG with $\mathbf{x}_k$ as query reaches $t$ exactly, from any entry point. This holds unconditionally whenever* sim *is Euclidean-derived, and more generally for any* self-dominant *kernel ($K(\mathbf{x}, \mathbf{x}) \geq K(\mathbf{y}, \mathbf{x})$ for all $\mathbf{x}, \mathbf{y}$ in the dataset).*

Corollary 1 follows purely from Lemma 4, an abstract statement about any graph with the no-close-disconnected-node property: nothing about SVG's own construction is used beyond the fact (Lemma 2) that it has this property. When $t \neq k$, only possible for non-metric similarities where self-similarity is not guaranteed maximal, $\mathcal{D}_k$ can contain points beyond $\{k, t\}$, and Lemma 4 only guarantees reaching *some* element of $\mathcal{D}_k$. Ruling this out requires going beyond the abstract graph property and back into the specific structure of the SVG's own edge-selection rule.

**Lemma 5** (SVG reciprocal-neighbor inclusion). *Let $k, t$ be distinct dataset indices with $K(\mathbf{x}_t, \mathbf{x}_k) > \max_{j \neq k, t} K(\mathbf{x}_t, \mathbf{x}_j)$ (i.e. $k$ is itself $\mathbf{x}_t$'s best match among all other points). Then $t \in \mathcal{N}_k$ in the SVG.*

Lemma 5, unlike Lemma 4, is not a fact about generalized monotonic search networks in the abstract: it is derived directly from the KKT stationarity of Problem (4), and would need to be re-derived from scratch for any other graph construction (e.g., MRNG and Vamana's own edge-selection rules do not obviously satisfy an analogous property). Combining it with Lemma 4 recovers an exact guarantee for a broader, still genuinely non-metric, regime.

**Assumption 1** (Mutual dominance). *For query $k$ with target $t := \operatorname{argmax}_i K(\mathbf{x}_i, \mathbf{x}_k)$, $t \neq k$:*

(a) $K(\mathbf{x}_i, \mathbf{x}_i) \leq K(\mathbf{x}_k, \mathbf{x}_k)$ *for every $i \notin \{k, t\}$;*
(b) $k = \operatorname{argmax}_{j \neq t} K(\mathbf{x}_t, \mathbf{x}_j)$ *($k$ and $t$ are mutual/reciprocal nearest neighbors).*

**Corollary 2** (Exact recovery under mutual dominance). *Under Assumption 1, Algorithm 1 on the SVG with $\mathbf{x}_k$ as query reaches $t$ exactly, from any entry point.*

Note the proof of Corollary 2 draws on both an abstract graph-theoretic fact (Lemma 4) and an SVG-specific one (Lemma 5); it is not a corollary of either alone.

Outside Assumption 1, Lemma 4 still guarantees convergence to $\mathcal{D}_k$, and empirically this coincides with exact recovery of $t$ overwhelmingly often even when the assumption is not checked: across 500-node subsamples of two real, non-normalized MIPS benchmarks (Netflix and Yahoo! Music, $t \neq k$ for the large majority of queries in both, see Table 3), greedy search on the SVG reached $t$ exactly in all 140,000 (query, entry-point) pairs tested (7 dataset/seed combinations); and on synthetic instances constructed to satisfy Assumption 1, all 15 tested instances reached $t$ exactly from every entry point, as the corollary predicts. A general, non-empirical characterization of how often Assumption 1 (or a weaker sufficient condition) holds on real embedding data is left to future work.

Indefinite kernels arise naturally for sequence data in time-series, protein, and genomics applications (e.g., Chen et al., 2009; Badiane & Cunningham, 2022). These kernels do not induce a RKHS and are not PSD. Most of the results in this section do not require $K$ to be PSD. Theorem 3, Lemma 2, the abstract dominance-set lemma Lemma 4, the self-dominant-kernel corollary Corollary 1, and the SVG-specific reciprocal-inclusion lemma Lemma 5 all rely only on the KKT conditions of Problem (7), which are meaningful whether or not

Table 1: Kernel assumptions behind our results. We indicate which results require a PSD kernel (inducing an RKHS) and which remain valid for indefinite kernels.

| Result | PSD / RKHS | Indefinite kernel |
|---|---|---|
| NNLS $\Leftrightarrow$ SVM interpretation; support-vector geometry (Section 3) | required | interpretation lost |
| Convex problem, unique solution, projected-gradient solver (Section 3.2) | required | non-convex |
| Navigability of SVG (Theorem 3) | not required | valid |
| Dominance-set convergence of Algorithm 1 (Lemma 4) | not required | valid |
| Exact recovery, self-dominant kernel (Corollary 1) | not required | valid |
| SVG reciprocal-neighbor inclusion (Lemma 5) | not required | valid |
| Exact recovery under mutual dominance (Corollary 2) | required | unavailable |
| SVG-L0 inherits guarantee when $\ell_0$ bound is inactive (Section 5) | not required | valid |
| Recovery of HNSW/MRNG/Vamana (Section 4) | RBF kernel | — |
| Sparsity bound ($\leq d+1$ out-edges, linear kernel) | required | — |

$K$ is PSD. Consequently, these results remain valid for indefinite kernels that do not induce an RKHS. Table 1 summarizes which of this section's results require a PSD kernel and which remain valid for indefinite kernels. With an indefinite kernel, Problem (7) is no longer convex and may admit multiple solutions; this does not affect the results above: since their guarantees depend only on the KKT conditions, which hold at every stationary point (local minimizer or saddle point), any such point yields the stated guarantee and the solver need only converge to one of them. Corollary 2 is the one exception: part (a) of Assumption 1 is verified via Cauchy–Schwarz in the RKHS induced by $K$, which presupposes an actual feature map $\phi$ with $K(\mathbf{x}, \mathbf{y}) = \phi(\mathbf{x})^\top \phi(\mathbf{y})$. For indefinite $K$ with $t \neq k$, Lemma 4 still guarantees convergence to $\mathcal{D}_k$, but this specific route to an exact guarantee is unavailable; recovering an analogous exact result for indefinite kernels, if one exists, is left to future work.

## 3.2 Computing the SVG

Writing Problem (7) in vectorial form, we obtain an instance of the problem

$$\min_{\mathbf{s}} \frac{1}{2}\mathbf{s}^\top \mathbf{A}\mathbf{s} - \mathbf{b}^\top \mathbf{s} \quad \text{s.t.} \quad \mathbf{s} \geq 0, \|\mathbf{s}\|_2^2 \leq n^{-1}, \tag{10}$$

where the PSD matrix $\mathbf{A}$ is result of removing the $i$-th row and the $i$-th column from the matrix $\mathbf{K}$ and $\mathbf{b}$ is result of removing the $i$-th element from $\mathbf{K}_{[:i]}$, the $i$-th column of $\mathbf{K}$. The feasible set $\mathcal{S} = \{\mathbf{s} : \mathbf{s} \geq \mathbf{0}, \|\mathbf{s}\|_2^2 \leq n^{-1}\}$ is convex, and its Euclidean projection is available in closed form: clip the negative entries to zero and, if the result leaves the ball, rescale it radially onto the sphere of radius $1/\sqrt{n}$,

$$\Pi_{\mathcal{S}}(\mathbf{y}) = \frac{\max\{\mathbf{y}, \mathbf{0}\}}{\max\{\sqrt{n}\,\|\max\{\mathbf{y}, \mathbf{0}\}\|_2,\, 1\}}. \tag{11}$$

This projection is exact: the nonnegativity multipliers give $s_j = \max\{y_j, 0\}/(1 + 2\nu)$ and the ball multiplier $\nu \geq 0$ sets the common scale. We therefore solve Problem (10) with accelerated projected gradient (FISTA), a single loop that alternates a gradient step on $F(\mathbf{s}) = \frac{1}{2}\mathbf{s}^\top \mathbf{A}\mathbf{s} - \mathbf{b}^\top \mathbf{s}$, whose gradient is $\mathbf{A}\mathbf{s} - \mathbf{b}$, with the projection in Equation (11):

$$\mathbf{s}_{t+1} = \Pi_{\mathcal{S}}\left(\mathbf{z}_t - \frac{1}{L}\left(\mathbf{A}\mathbf{z}_t - \mathbf{b}\right)\right), \qquad \mathbf{z}_{t+1} = \mathbf{s}_{t+1} + \frac{\theta_t - 1}{\theta_{t+1}}\left(\mathbf{s}_{t+1} - \mathbf{s}_t\right), \tag{12}$$

with $\theta_{t+1} = \frac{1}{2}\left(1 + \sqrt{1 + 4\theta_t^2}\right)$ and step size $1/L$, where $L$ is the largest eigenvalue of $\mathbf{A}$ (cheaply upper-bounded by a few power iterations, or by the maximum row sum of $\mathbf{A}$). Because the projection is part of the iteration, the fixed points of Equation (12) are exactly the KKT points of the constrained Problem (10); the method converges to its global minimizer and the constraints hold exactly at every iterate.

Each FISTA iteration is dominated by the matrix-vector product $\mathbf{A}\mathbf{z}_t$ at cost $O(n^2)$, uses only matrix-vector products and elementwise operations (efficient on GPUs or via AVX on x86 CPUs), and converges in a number of iterations that grows slowly with $n$ (Figure 6, left). Compared against the general-purpose convex

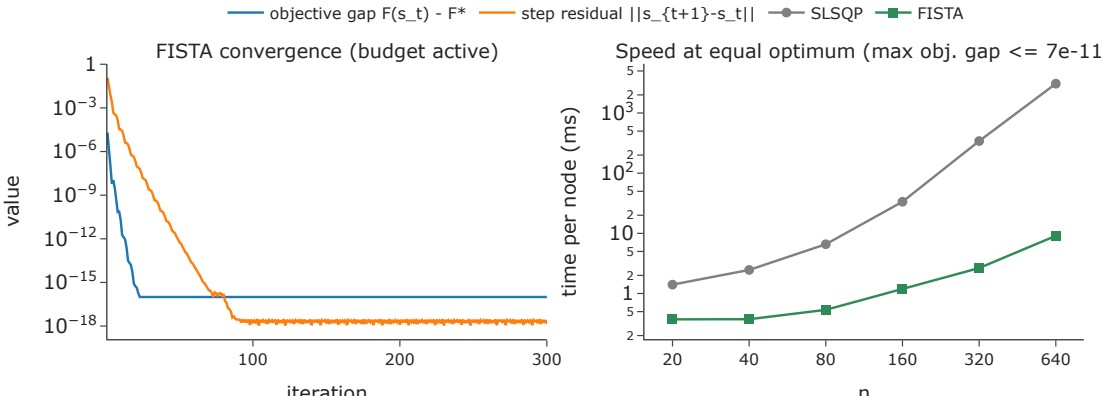

Figure 6: Empirical behavior of the projected-gradient (FISTA) solver in Equation (12) for Problem (10) with an RBF kernel. **Left:** on a representative node where the budget is active, the objective gap to the optimum $F(\mathbf{s}_t) - F^\star$ (blue) and the step residual $\|\mathbf{s}_{t+1} - \mathbf{s}_t\|_2$ (orange) decay rapidly, reaching the optimum to machine precision within a few tens of iterations. **Right:** wall-clock time per node as a function of $n$ for FISTA and the general-purpose convex solver SLSQP (scipy), both converged to the same optimum (objective gap below $10^{-13}$). FISTA is comparable at the smallest sizes and scales far better as $n$ grows, since each iteration costs a single matrix-vector product ($O(n^2)$) rather than a dense factorization ($O(n^3)$).

---

**Algorithm 3:** Pruning meta-algorithm to determine the outgoing edges for node $i$

---

**Input**    : Dataset $\mathcal{X} = \left\{\mathbf{x}_j \in \mathbb{R}^d\right\}_{j=1}^n$, node $i \in [1 \dots n]$, candidate pool $\mathcal{C}_i$.
**Output:** Set $\mathcal{N}_i$ of outgoing neighbors for node $i$
**1** $\mathcal{E}_i \leftarrow \emptyset$;
**2 while** $\mathcal{C}_i \neq \emptyset$ **do**
**3**   $\quad j \leftarrow \underset{j' \in \mathcal{C}}{\operatorname{argmax}} K(\mathbf{x}_i, \mathbf{x}_{j'})$;
**4**   $\quad \mathcal{N}_i \leftarrow \mathcal{E}_i \cup \{j\}$;
**5**   $\quad \mathcal{C}_i \leftarrow \mathcal{C}_i \setminus \{j\}$;
**6**   $\quad \mathcal{C}_i \leftarrow \{k \in \mathcal{C}_i \,|\, \text{connectivity rule between } i, j, \text{ and } k \text{ is met}\}$;

---

solver SLSQP (scipy), which reaches the same optimum, FISTA is comparable on the smallest problems and increasingly faster as $n$ grows (matrix-vector iterations of cost $O(n^2)$ versus dense factorizations of cost $O(n^3)$ per iteration), exceeding an order of magnitude speedup by $n \approx 100$ (Figure 6, right).

With an $O(n^2)$ cost per node, the full SVG runs in $O(n^3)$. This matches the cost of the degree-unconstrained versions of MRNG (Fu et al., 2019), HNSW (Malkov & Yashunin, 2020), and Vamana (Subramanya et al., 2019). Among these graph indices, only SVG carries a navigability guarantee for general, including non-metric, similarities; MRNG, HNSW, and Vamana are guaranteed navigable only in the Euclidean setting. The full SVG is therefore primarily an analytical object; for practical, bounded-degree construction we use SVG-L0 (Section 5), which lowers the per-node cost to $O(M^2)$.

## 4 Connecting SVG to other graph indices

We now study the link between SVG and other popular graph indices. Algorithm 3 provides a blueprint for most pruning techniques (Malkov & Yashunin, 2020; Subramanya et al., 2019; Fu et al., 2022) used in Algorithm 2. Given a candidate pool $\mathcal{C}_i$, Algorithm 3 considers triplets of nodes $i, j, k$, as depicted in Figure 7. Throughout this section, we use the standard assumption $\mathcal{C}_i = [1 \dots n] \setminus \{i\}$ (we discuss this choice in the next section).

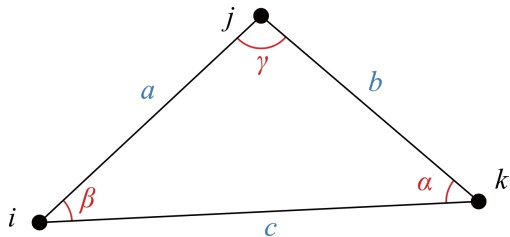

Figure 7: We show that, when using an RBF kernel, the graph pruning rules of most popular graph construction algorithms, including the popular HNSW (Malkov & Yashunin, 2020) and DiskANN (Subramanya et al., 2019), can be written as applications of the law of cosines and the inequality $a^2 + b^2 > c^2$.

Next, we show that Problem (4), when applied to the analysis of these triplets, leads to a traditional graph sparsification algorithm with navigability guarantees for general kernels. Given a PSD kernel $K$, for each triplet $i, j, k$ encountered in Line 6 of Algorithm 3, we solve Problem (4) using just $\phi(\mathbf{x}_i)$, $\phi(\mathbf{x}_j)$, and $\phi(\mathbf{x}_k)$ and connect $i$ to $j$ and/or $k$ if $s_i > 0$ and/or $s_k > 0$.

We say that a kernel is normalized if $(\forall \mathbf{x}) K(\mathbf{x}, \mathbf{x}) = 1$. Using Proposition 2 in the appendix, we derive the following equivalence for normalized kernels, which leads to a monotonicity certificate.

**Lemma 6** (Kernel connectivity rule). *Let $K$ be a normalized PSD kernel. Solving Problem (4) for each triplet $i, j, k$ encountered in Line 6 of Algorithm 3 is equivalent to $k \in \mathcal{N}_i$ if and only if*

$$K(\mathbf{x}_i, \mathbf{x}_j) K(\mathbf{x}_j, \mathbf{x}_k) < K(\mathbf{x}_i, \mathbf{x}_k). \tag{13}$$

**Theorem 4.** *Let $s, k$ be two distinct nodes of a graph built using Algorithm 3 with $\mathcal{C}_i = [1 \ldots n] \setminus \{i\}$ and the kernel connectivity rule in Lemma 6 and let $t = \underset{i \in V}{\operatorname{argmax}} K(\mathbf{x}_i, \mathbf{x}_k)$. There is a generalized monotonic path between $s$ and $t$.*

The following corollary follows immediately from Definition 2 and Theorem 4.

**Corollary 3.** *A graph built using Algorithm 3 with $(\forall i) \mathcal{C}_i = [1 \ldots n]$ and the connectivity rule in Lemma 6 is a generalized monotonic search network.*

This is the second graph construction algorithm with full navigability guarantees regardless of the similarity function underlying the kernel (SVG is the first). In this sense, it applies to any metric space and even non-metric vector spaces, e.g., spaces only equipped with an inner product.

For the RBF kernel, the connectivity rule in Lemma 6 amounts to the familiar inequality $a^2 + b^2 > c^2$, as depicted in Figure 7 and shown next.

**Corollary 4.** *When using the RBF kernel in the same setting as Proposition 2, we have*

$$K(\mathbf{x}_i, \mathbf{x}_j) K(\mathbf{x}_j, \mathbf{x}_k) < K(\mathbf{x}_i, \mathbf{x}_k) \quad \Leftrightarrow \quad \|\mathbf{x}_i - \mathbf{x}_j\|_2^2 + \|\mathbf{x}_j - \mathbf{x}_k\|_2^2 > \|\mathbf{x}_i - \mathbf{x}_k\|_2^2. \tag{14}$$

From Corollary 4, we can derive many graph construction rules by combining $a^2 + b^2 > c^2$ with different formulas from the law of cosines. Corollaries 5 and 6 provide two specific examples (in the following, we connect them with existing graph indices).

**Corollary 5** (Shekkizhar & Ortega, 2023). *When using an RBF kernel in the same setting as Lemma 6, the necessary condition to connect node $i$ with node $k$ is*

$$(\cos \alpha) \|\mathbf{x}_i - \mathbf{x}_k\|_2 < \|\mathbf{x}_j - \mathbf{x}_k\|_2, \tag{15}$$

*where $\alpha$ is the angle between the vectors $\mathbf{x}_i - \mathbf{x}_k$ and $\mathbf{x}_j - \mathbf{x}_k$.*

**Corollary 6.** *When using an RBF kernel in the same setting as Lemma 6, the necessary condition to connect node $i$ with node $k$ is*

$$(\cos \beta) \|\mathbf{x}_i - \mathbf{x}_k\|_2 < \|\mathbf{x}_i - \mathbf{x}_j\|_2, \tag{16}$$

*where $\beta$ is the angle between the vectors $\mathbf{x}_k - \mathbf{x}_i$ and $\mathbf{x}_j - \mathbf{x}_i$.*

### 4.1 Graph indices under the lens

In the following, we show that the most popular graph indices can be interpreted as thresholded applications of different formulas from the law of cosines combined with $a^2 + b^2 > c^2$ (see Figure 7). In particular, our results cover the popular HNSW (Malkov & Yashunin, 2020) and DiskANN (Subramanya et al., 2019).

We start with the connectivity rule shared by MRNG (Fu et al., 2019) and HNSW (Malkov & Yashunin, 2020). For $i, j \in [1 \ldots n]$, let

$$lune_{ij} = \left\{ \mathbf{x} \in \mathbb{R}^d \mid \|\mathbf{x}_i - \mathbf{x}\|_2 \leq \|\mathbf{x}_i - \mathbf{x}_j\|_2 \wedge \|\mathbf{x}_j - \mathbf{x}\|_2 \leq \|\mathbf{x}_i - \mathbf{x}_j\|_2 \right\} \tag{17}$$

An MRNG is a directed graph $G = (\mathcal{V}, \mathcal{E})$ with $\mathcal{V} = [1 \ldots n]$ and the edge set $\mathcal{E}$ satisfying the following property: For any pair $i, k \in [1 \ldots n]$, $\overrightarrow{ik} \in \mathcal{E}$ if and only if $lune_{ik} \cap S = \emptyset$ or $(\forall r) \, \mathbf{x}_r \in lune_{ik} \cap S \implies \overrightarrow{ir} \notin \mathcal{E}$. MRNG and HNSW use the following connectivity rule in Algorithm 3: Keep $k$ in $\mathcal{C}_i$ if

$$\|\mathbf{x}_i - \mathbf{x}_k\|_2 \leq \|\mathbf{x}_j - \mathbf{x}_k\|_2 . \tag{18}$$

HNSW applies this rule within a hierarchical structure. A direct application of Corollary 5 yields the following result.

**Corollary 7.** *Running Algorithm 3 with the RBF kernel and the MRNG connectivity rule in Equation (18) is equivalent to applying the necessary condition in Corollary 5 with the additional simplification that $\cos \alpha = 1$.*

Vamana, the algorithm behind DiskANN (Subramanya et al., 2019), is an extension of MRNG (Fu et al., 2019) that seeks to accelerate the graph traversal speed. Recently, this acceleration has been formally proven in the worst case (Indyk & Xu, 2023). Vamana uses the following connectivity rule in Algorithm 3: keep $k$ in $\mathcal{C}_i$ if

$$\|\mathbf{x}_i - \mathbf{x}_k\|_2 \leq \lambda \|\mathbf{x}_j - \mathbf{x}_k\|_2 . \tag{19}$$

A direct application of Corollary 5 yields the following result.

**Corollary 8.** *Running Algorithm 3 with the RBF kernel and the Vamana connectivity rule in Equation (19) is equivalent to applying the necessary condition in Corollary 5 with the additional simplification that $\alpha = \arccos \lambda^{-1}$.*

We also extend these results to the recently introduced SSG (Fu et al., 2022) using the following definitions

$$\text{angle}(\mathbf{x} \,, \mathbf{y}) = \arccos \frac{\langle \mathbf{x} \,, \mathbf{y} \rangle}{\|\mathbf{x}\|_2 \|\mathbf{y}\|_2}, \tag{20}$$

$$\text{ball}(i, \delta) = \left\{ \mathbf{x} \in \mathbb{R}^d \mid \|\mathbf{x} - \mathbf{x}_i\|_2 \leq \delta \right\}, \tag{21}$$

$$\text{cone}_{ij}^\theta = \left\{ \mathbf{x} \in \mathbb{R}^d \mid \text{angle}(\mathbf{x} - \mathbf{x}_i \,, \mathbf{x}_j - \mathbf{x}_i) \leq \theta \right\}. \tag{22}$$

An SSG is a directed graph $G = (\mathcal{V}, \mathcal{E})$ with $\mathcal{V} = [1 \ldots n]$ and the edge set $\mathcal{E}$ satisfying the following property: For any pair $i, k \in [1 \ldots n]$, $\overrightarrow{ik} \in \mathcal{E}$ if and only if $\text{cone}_{ik}^\theta \cap \text{ball}(i, \|\mathbf{x}_i - \mathbf{x}_k\|_2) \cap S = \emptyset$ or $(\forall r, \mathbf{x}_r \in \text{cone}_{ik}^\theta \cap \text{ball}(i, \|\mathbf{x}_i - \mathbf{x}_k\|_2) \cap S) \, \overrightarrow{ir} \notin \mathcal{E}$, where $0 \leq \theta \leq 60°$ is a hyperparameter. SSG uses the following connectivity rule in Algorithm 3: for $0 \leq \theta \leq 60°$, keep $k$ in $\mathcal{C}_i$ if

$$\text{angle}(\mathbf{x}_j - \mathbf{x}_i \,, \mathbf{x}_k - \mathbf{x}_i) \geq \theta \vee \|\mathbf{x}_j - \mathbf{x}_i\|_2 \geq \|\mathbf{x}_i - \mathbf{x}_k\|_2 . \tag{23}$$

A direct application of Corollary 6 yields the following result.

**Corollary 9.** *Running Algorithm 3 with the RBF kernel and the SSG connectivity rule in Equation (23) rule is equivalent to applying the necessary condition in Corollary 6 with the additional simplification that $\theta = \arccos \left( \|\mathbf{x}_i - \mathbf{x}_j\|_2 / \|\mathbf{x}_i - \mathbf{x}_k\|_2 \right)$.*

In summary, the edge pruning rules used in Algorithm 3 by some of the most popular graph indices can be regarded as specializations of the SVG optimization. These specializations, described in Figure 7, emerge from applying the optimization to triplets of points.

---

**Algorithm 4:** Subspace pursuit for SVG construction

---

    **Input**  : Dataset $\mathcal{X} = \left\{\mathbf{x}_j \in \mathbb{R}^d\right\}_{j=1}^n$, element $i \in [1\ldots n]$, maximum out-degree $M \in \mathbb{N}^+$.

    **Output:** Set $\mathcal{N}_i$ of outgoing neighbors for node $i$.

**1** Select the kernel width $\sigma$ used for node $i$;

**2** $\mathcal{N}^{(0)} \leftarrow \emptyset$;

**3 for** $t \in [1\ldots T]$ **do**

**4**      Let $\mathcal{C}_i$ be the set of the $M$ largest entries in $\left\{K(\mathbf{x}_i, \mathbf{x}_k) - \sum_{j \in \mathcal{N}^{(t-1)}} s_j^{(t-1)} K(\mathbf{x}_j, \mathbf{x}_k) \,\middle|\, k \in [1\ldots n]\right\}$;

**5**      $\mathcal{C}_i \leftarrow \mathcal{C}_i \cup \mathcal{N}^{(t-1)}$;

**6**      Find the solution $\mathbf{s}$ to Problem (24), restricted to vectors in $\mathcal{C}_i$;

**7**      Let $\mathcal{N}^{(t)}$ be the set of indices corresponding to the $M$ largest entries of $\mathbf{s}$;

**8**      **if** $\mathcal{N}^{(t)} = \mathcal{N}^{(t-1)}$ **then** break;

**9** $\mathcal{N}_i \leftarrow \mathcal{N}^{(t)}$;

---

## 5 Fast SVG construction with bounded out-degree

Graph construction algorithms based on Algorithm 3, such as those described in Section 4, have two main practical issues that require careful tuning.

First, Algorithm 3 requires a candidate pool $\mathcal{C}_i$. Setting $\mathcal{C}_i = [1\ldots n] \setminus \{i\}$ is usually used to obtain theoretical guarantees, as described in Section 4, but becomes practically untenable as $n$ grows. In practice, $\mathcal{C}_i$ is heuristically determined by finding the (approximate) nearest neighbors of $\mathbf{x}_i$. However, this may be problematic if, for example, $\mathbf{x}_i$ lies on the outskirts of a tight cluster (Indyk & Xu, 2023) as the graph may become disconnected. Take the example of the red point in Figure 9. We would need to create a candidate pool larger than the number of points in the left cluster for Algorithm 3 to ensure navigability between both clusters.[3] Finding a prudent size for $\mathcal{C}_i$ becomes a dataset-specific tuning problem.

Second, although Algorithm 3 produces sparse graphs when paired with a suitable edge selection rule (such as those in Section 4), the graphs are generally not sparse enough. The sparsity of these graphs is critical as it directly determines its footprint and search runtime. Let $M$ be the maximum out-degree we want in a graph. Because Algorithm 3 neither produces a total order nor handles the cardinality constraint intrinsically, the list of neighbors is truncated in an ad-hoc fashion by stopping Algorithm 3 once $|\mathcal{N}_i| = M$. This heuristic may cause navigability problems, as described in Figure 8. In essence, the diversity of the selected edges becomes suboptimal and may prevent navigation if the process is terminated early.

In this section, we show that the SVG framework overcomes these difficulties. Although the solution to Problem (4) is naturally sparse, we would like to impose a more stringent and specific level of sparsity to bound the out-degree of the graph. Moreover, we show that once this restriction is added, precomputing a candidate pool becomes unnecessary.

We address both problems simultaneously by altering the SVG optimization. We bound the sparsity level by solving the related problem

$$\min_{\mathbf{s}} \frac{1}{2} \|\phi(\mathbf{x}_i) - \boldsymbol{\Phi}\mathbf{s}\|_2^2 \quad \text{s.t.} \quad \mathbf{s} \geq \mathbf{0},\ s_i = 0, \|\mathbf{s}\|_2^2 \leq n^{-1}, \|\mathbf{s}\|_0 \leq M, \tag{24}$$

where the so-called $\ell_0$ norm measures the number of non-zero entries of a vector. We use Problem (24) to build a graph with a maximum out-degree, since the minimizer $\mathbf{s}_i^*$ will have at most $M$ nonzero entries. In contrast with the typical truncation heuristic, Problem (24) selects the subset of $M$ elements that provides the best tradeoff between diversity and similarity (see the analysis of Problem (7)). This approach has connections with sparse SVMs (Smola et al., 1999), which use parsimony-inducing $\ell_1$ (e.g., Bi et al., 2003) or $\ell_0$ (e.g., Zhang et al., 2023) constraints to sparsify the support vector set.

---

[3]The graph construction by Shekkizhar & Ortega (2023) that uses Problem (4) for manifold learning shares these issues.

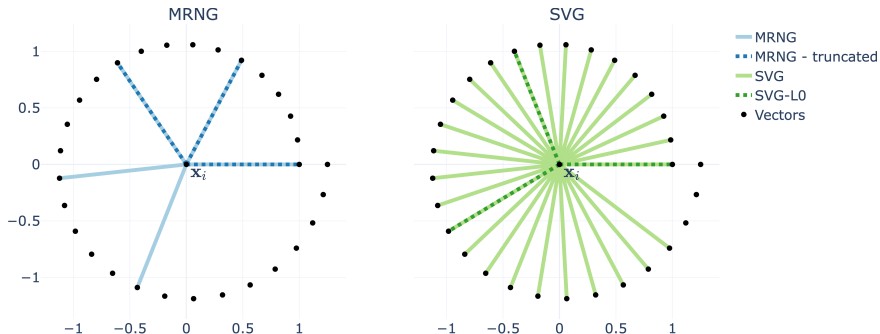

Figure 8: The MRNG orders a node's candidate out-edges by increasing distance and keeps those not occluded by a closer neighbor. Bounding the degree by truncating this list once $\mathcal{N}_i$ reaches a prescribed size therefore retains the nearest occlusion survivors and discards farther, more diverse neighbors; with the resulting edge set (dotted blue lines, left), navigating downwards from $\mathbf{x}_i$ is not possible. SVG-L0 (Problem (24)) instead selects, at the same out-degree, the subset that best solves the cardinality-constrained problem. Because its objective balances similarity against diversity, it trades a redundant near neighbor for one covering the missing direction (dotted green lines, right), restoring navigability. This gain is most pronounced when the competing neighbors have comparable similarity to $\mathbf{x}_i$, as is typical in the high-dimensional regime of Section 6. The SVG-L0 edges are not necessarily a subset of the SVG edges.

The astute reader will notice that solving Problems (4) and (24) quickly becomes impractical as $n$ grows: The size of the kernel matrix $\mathbf{K}$ is quadratic in $n$ and we solve an optimization with $n$ variables. It seems like our computational requirements are still very high. Additionally, problems involving $\ell_0$ constraints have always been considered challenging because of their non-convexity and NP-hardness. The dominant paradigm replaces these constraints by convex $\ell_1$ constraints (e.g., Candès & Plan, 2009).[4] However, Problem (24) belongs to a particular family of problems, known as subspace pursuit, for which there are very efficient algorithms (Dai & Milenkovic, 2009; Needell & Tropp, 2010). Algorithm 4 presents an algorithm that solves this problem. Additional details on subspace pursuit can be found in Appendix C.

Algorithm 4 does not depend on precomputing an appropriate candidate pool, in contrast to Algorithm 2. This superpower comes from Line 4, which performs a neighbor search with a modified similarity. By finding the vectors that maximize

$$K(\mathbf{x}_i, \mathbf{x}_k) - \sum_{j \in \mathcal{N}^{(t-1)}} s_j^{(t-1)} K(\mathbf{x}_j, \mathbf{x}_k), \tag{25}$$

it becomes clear that the similarity in this neighbor search selects vectors that are close to $\mathbf{x}_i$ and far away from the vectors in $\mathcal{N}^{(t-1)}$. This search focuses its attention on portions of the space not considered in previous iterations (see Figure 9).

As a side note, by writing Equation (25) as

$$\mathbf{v}^\top \phi(\mathbf{x}_k), \quad \text{where} \quad \mathbf{v} = \phi(\mathbf{x}_i) - \sum_{j \in \mathcal{N}^{(t-1)}} s_j^{(t-1)} \phi(\mathbf{x}_j), \tag{26}$$

the computation can be carried out using random features (RF) (Rahimi & Recht, 2007; Reid et al., 2023; Liu et al., 2022; Sernau et al., 2024). However, the literature has understandably concentrated on approximating the central portion of kernels instead of their tails (e.g., for exponential kernels, where $K_{\text{EXP}}(\mathbf{x}, \mathbf{x}') \approx 0_+$). Since Equation (26) is concerned with the tails (notice the small values in the attention area in Figure 9), new RF techniques would be needed. This is an interesting future line of work.

**Computational complexity.** Algorithm 4 involves solving a least squares problems on the simplex in Line 6 with $2M$ variables ($M$ from $\mathcal{C}_i$ and $M$ from $\mathcal{N}^{(t-1)}$). Thus, the complexity of the sub-problem is drastically reduced from $O(n^2)$ to $O(M^2)$ when using the projected-gradient solver of Section 3.2. Implementing

---

[4]Direct $\ell_0$ solvers have gained significant attention in the last few years (Bertsimas et al., 2016; Hastie et al., 2020) due to improvements in mixed integer optimization.

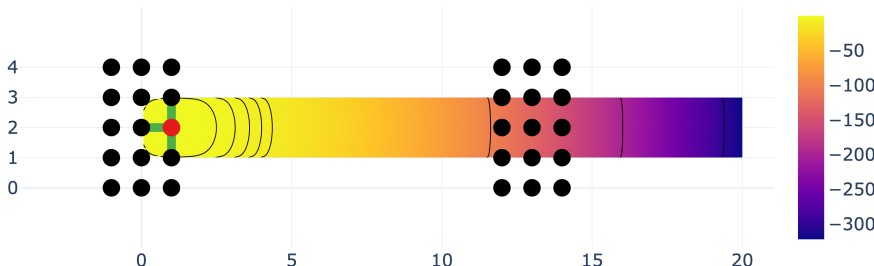

Figure 9: The attention of the search driven by Equation (25) after connecting the red point to its three neighbors (green edges). We show the values of Equation (25) in a logarithmic colorscale in the area where it is positive. The attention will focus on finding the next $M$ points to the right of the red point, seeking to find another support vector for the SVM classifier.

Line 4 in Algorithm 4 using a vector search index that typically offer $O(\log n)$ complexity, the complexity of Algorithm 4 becomes sublinear in the number of indexed vectors. When running Algorithm 4 for each node in the graph, the total SVG-L0 construction complexity is $O(n(M^2 + \log n))$. This complexity, as function of $n$, is similar to other existing graph indices (Malkov & Yashunin, 2020; Fu et al., 2019; Subramanya et al., 2019).

## 5.1 Towards a characterization of SVG-L0

We now make explicit the relationship between SVG-L0 and the navigability guarantee of Theorem 3. When the SVG solution already satisfies the degree bound (i.e., $\|\mathbf{s}^*\|_0 \leq M$), the $\ell_0$ constraint is inactive and Problem (24) shares its minimizer with the SVG; in this regime, SVG-L0 inherits the full navigability guarantee. This holds for a broad class of kernels: whenever the features $\phi(\mathbf{x})$ lie in a $D$-dimensional space, the SVG has at most $D + 1$ out-edges per node (the SVM support-vector bound; Section 3), so any $M \geq D+1$ yields a fully navigable graph. This covers the linear/MIP kernel ($D = d$), polynomial kernels, and any explicit finite-dimensional feature map. Notably, in the MIP setting SVG-L0 thus provides a navigable graph with at most $d+1$ out-edges per node, in contrast to the inner-product Delaunay graph (Section 3.1), which is navigable but whose number of edges grows exponentially with $d$ (Morozov & Babenko, 2018).

Many widely used kernels, however, have infinite-dimensional features, including the exponential/RBF kernel central to this paper; for these, no finite $M$ guarantees inheritance a priori, and the out-degree of the SVG depends on the data geometry (Figure 5) rather than the feature dimension. We therefore do not rest the general argument on this bound: it identifies a clean regime where SVG-L0 provably inherits navigability. When the $\ell_0$ bound binds, SVG-L0 no longer inherits the exact guarantee: a hard out-degree bound precludes full navigability for arbitrary data. We can nonetheless characterize what navigability requires in this regime, and identify the quantity that governs it. Recall (Lemma 2) that greedy search progresses from $\mathbf{x}_i$ toward a target $\mathbf{x}_t$ iff some retained neighbor $j$ *occludes* $t$, i.e. $K(\mathbf{x}_j, \mathbf{x}_t) > K(\mathbf{x}_i, \mathbf{x}_t)$, equivalently $\phi(\mathbf{x}_t)^\top(\phi(\mathbf{x}_j) - \phi(\mathbf{x}_i)) > 0$.

**Definition 8** (Occlusion cover and covering number). *For a similarity floor $\beta \geq 0$, the out-neighborhood $\mathcal{N}_i$ is an* occlusion cover *if every $t \neq i$ with $K(\mathbf{x}_i, \mathbf{x}_t) \geq \beta$ is occluded by some $j \in \mathcal{N}_i$. The* occlusion covering number $C_\beta$ *is the smallest out-degree for which every node admits an occlusion cover.*

**Proposition 1.** *Fix $\beta \geq 0$. Suppose the directed graph $G = ([1 \ldots n], \mathcal{E})$ is an occlusion cover at level $\beta$: for every node $i$ and every query index $k$ with $K(\mathbf{x}_i, \mathbf{x}_k) \geq \beta$ and $i \neq t_k$, where $t_k := \arg\max_j K(\mathbf{x}_j, \mathbf{x}_k)$, there exists $j \in \mathcal{N}_i$ with $K(\mathbf{x}_j, \mathbf{x}_k) > K(\mathbf{x}_i, \mathbf{x}_k)$. Then for every query $\mathbf{x}_k$ and every entry point $i$ with $K(\mathbf{x}_i, \mathbf{x}_k) \geq \beta$, Algorithm 1 reaches $t_k$. In particular, a navigable graph of out-degree $C_\beta$ (Definition 8) exists.*

The covering number $C_\beta$ is the quantity that governs binding-regime navigability and behaves very differently from the exact-SVG degree: $C_\beta$ is a property of the data geometry, scaling with the intrinsic dimension and *independent of $n$*, whereas the SVG can use up to $n-1$ support vectors. For metric kernels, where occlusion

reduces to "$\mathbf{x}_j$ closer to $\mathbf{x}_t$ than $\mathbf{x}_i$", $C_\beta$ is bounded by a doubling-dimension degree bound in the spirit of navigating nets and relative neighborhood graphs (e.g., Krauthgamer & Lee, 2004).

We point out that the occlusion cover is a characterization rather than a construction: like the SVG and the MRNG/Delaunay graphs, realizing the occlusion cover requires the full pairwise occlusion structure (a candidate pool) and, in general, the solution of a combinatorial cover, precisely the costs that motivate SVG-L0 (Section 5). Its role is to identify the governing quantity $C_\beta$ and the achievable optimum; SVG-L0 is the actionable, candidate-pool-free construction that approximates it at $O(M^2)$ per node.

We now show that SVG-L0's construction is aligned with this requirement: we show that any not-yet-occluded target within similarity $\beta$ has residual similarity at least $\beta/2$. Recall that the candidate step of Algorithm 4 selects the point of largest residual similarity (Equation (25)),

**Lemma 7.** *Let $\mathbf{s}^*$ be a minimizer of Problem (24) at node $i$ (so $\mathbf{s}^* \geq \mathbf{0}$, $\|\mathbf{s}^*\|_2^2 \leq n^{-1}$, $\|\mathbf{s}^*\|_0 \leq M$), with out-neighbors $\mathcal{N}_i = \mathrm{supp}(\mathbf{s}^*)$ and residual $\mathbf{r} = \phi(\mathbf{x}_i) - \mathbf{\Phi}\mathbf{s}^*$. Assume $M \leq n/4$. Then for every point $t$ with $K(\mathbf{x}_i, \mathbf{x}_t) \geq \beta$ that is* not *occluded by $\mathcal{N}_i$ (i.e. $K(\mathbf{x}_j, \mathbf{x}_t) \leq K(\mathbf{x}_i, \mathbf{x}_t)$ for all $j \in \mathcal{N}_i$),*

$$\mathbf{r}^\top \phi(\mathbf{x}_t) \;=\; K(\mathbf{x}_i, \mathbf{x}_t) - \sum_{j \in \mathcal{N}_i} s_j^* K(\mathbf{x}_j, \mathbf{x}_t) \;\geq\; \tfrac{1}{2} K(\mathbf{x}_i, \mathbf{x}_t) \;\geq\; \tfrac{\beta}{2}.$$

*Consequently, while any $\beta$-relevant target is unoccluded, the candidate selected by Equation (25) (which maximizes $\mathbf{r}^\top \phi(\mathbf{x}_a) = K(\mathbf{x}_i, \mathbf{x}_a) - \sum_j s_j^* K(\mathbf{x}_j, \mathbf{x}_a)$) has residual similarity at least $\beta/2$: the SVG-L0 is directed at uncovered targets.*

Empirically, SVG-L0 attains navigability at an out-degree that is a small multiple of $C_\beta$ across both metric (RBF) and non-metric (inner-product, $t \neq k$) similarities (Section 6), and it improves over the truncated MRNG precisely when covering is non-trivial. A closed-form bound on this out-degree multiple for the SVG-L0 minimizer is left to future work.

# 6 Experimental results

We present a few experimental results to highlight the practical value of SVG and SVG-L0. For most experiments and unless specified, we use the standard recall@1 measure of search accuracy, which counts how many times we find the ground truth NN of every vector $\mathbf{x}_i \in \mathcal{X}$ when using it as the query. In addition to purely greedy graph search algorithms, we also experimented with backtracking since it is widely used in practice (see Algorithm 6 in the appendix). We use small backtracking queues of length 2, 5, and 10.

Our results in Section 3.1 predict the navigability of the SVG. This is empirically verified by the experiment in Figure 10, where the empirical navigability (recall@1) is constant and perfect.

We also compared SVG-L0 (Section 5) with the degree-constrained MRNG, the truncated MRNG, and the truncated Vamana. As discussed in Section 4, the MRNG/Vamana are a fully navigable network with no degree constraints. This feature comes at a steep price: the complexity of building with MRNG/Vamana is $O(n^3)$. Here, we use a (still computationally costly) variant that uses a maximum out-degree $M$ and an unlimited candidate pool size, resulting in a complexity of $O(n^2)$. The truncated MRNG/Vamana have the additional constraint of working with a fixed candidate pool size (see Algorithm 2). Note that the truncated MRNG/Vamana are the algorithms used in practice today to build graph indices. For the baseline indices, we set the candidate pool size as a multiplicative factor of $M$, that is, $|\mathcal{C}| = rM$ for $r > 1$, as is standard practice. SVG-L0 uses a maximum out-degree but does not need a candidate pool.

Table 2 verifies that binding-regime navigability (Section 5.1) is governed by the occlusion covering number $C_\beta$, and that SVG-L0 operates close to it, on two real datasets: a metric one (ada-002 embeddings with the RBF kernel, $t = k$) and a non-metric one (Netflix inner-product embeddings, $t \neq k$ for 94% of queries), using recall@1 under greedy search with short backtracking. In both cases the occlusion covering number $C_\beta$ is small and essentially independent of $n$ ($C_\beta \approx 10$ for ada-002 and $C_\beta \approx 2$ for Netflix, stable from $n = 800$ to 2000), unlike the exact-SVG degree, which grows with $n$. SVG-L0 attains near-perfect navigability at an out-degree only a few times $C_\beta$ for both similarities; for example, recall@1 $\approx 0.99$ at $M \approx 3\,C_\beta$ on ada-002.

Table 2: Binding-regime navigability against the occlusion covering number $C_\beta$. On two real datasets ($n = 2000$) we report $C_\beta$ (greedy estimate; maximum over nodes in parentheses) and recall@1 (greedy search with short backtracking) for SVG-L0 and the truncated MRNG at out-degrees $M$. SVG-L0 reaches near-perfect recall at $M$ a small multiple of $C_\beta$ for both a metric (RBF, $t = k$)and a non-metric (inner product, $t \neq k$ for 94% of queries) similarity. It improves over the truncated MRNG when covering is non-trivial (RBF) and matches it when covering is trivial ($C_\beta \approx 2$, MIP), as the covering characterization predicts. $C_\beta$ is small and stable in $n$ ($9.1, 9.1, 9.9$ for ada-002 and $1.4, 1.5, 2.0$ for Netflix at $n = 800, 1500, 2000$).

|  |  | $M = 8$ | | $M = 16$ | | $M = 32$ | |
| --- | --- | --- | --- | --- | --- | --- | --- |
| Dataset (kernel) | $C_\beta$ (max) | SVG-L0 | trunc. MRNG | SVG-L0 | trunc. MRNG | SVG-L0 | trunc. MRNG |
| ada-002 (RBF, $t = k$) | 9.9 (16) | 0.636 | 0.592 | 0.909 | 0.883 | 0.986 | 0.977 |
| Netflix (MIP, $t \neq k$) | 2.0 (12) | 0.849 | 0.842 | 0.969 | 0.962 | 0.987 | 0.987 |

Figure 10: As Theorem 3 predicts, SVGs are navigable graphs in different dimensions and independently of the value of $\sigma$. We compute the reachability of Algorithm 1 for every pair of source and target vectors using recall@1 over ten realizations of 500 random vectors with different numbers of dimensions.

Finally, at matched out-degree SVG-L0 improves over the truncated MRNG exactly when covering is non-trivial (the RBF case, consistent with Figure 13) and coincides with it when covering is trivial ($C_\beta \approx 2$, the MIP case), precisely as the covering characterization predicts.

As shown in Figure 11, SVG-L0 is competitive with the degree-constrained MRNG when working with randomly distributed vectors. Its accuracy is slightly lower in the greedy setting, but slightly higher in the backtracking setting. Both indices are clearly superior to the truncated MRNG and the truncated Vamana.

The same synthetic experiment can be repeated under inner-product similarity to compare SVG-L0 against ip-NSW (Morozov & Babenko, 2018), its natural non-metric counterpart, since (i) ip-NSW's practical construction is exactly the un-diversified top-$M$ rule that SVG's candidate step generalizes and (ii) its theoretical underpinning competes directly with SVG (Section 3.1). As shown in Figure 12, SVG-L0 outperforms ip-NSW at every dimension and backtracking budget tested, with the margin widening as backtracking increases, consistent with ip-NSW's construction performing no pruning or diversification at all.

Lastly, we experimented with a few small real-world datasets using Python implementations of SVG and SVG-L0 that were not optimized for scaling. We took $10^4$ vectors from the datasets in Table 3 of the appendix with $d = 128, 1024, 1536, 3072$ dimensions. We built indices with SVG-L0, with the truncated MRNG described previously, and with the truncated Vamana. We observe in Figure 13 that SVG-L0 outperforms the truncated MRNG and the truncated Vamana. The effect is more pronounced in higher dimensions. In all dimensions, the value of $\sigma$ in SVG-L0 has little effect on its accuracy. We include a detailed analysis of the least favorable and a more favorable examples to SVG in Figure 14.

The scale of $\sigma$ that yields good navigability does grow with the dimension $d$ (Figure 10), which is expected rather than a sign of fragility: as $d$ grows, inter-point distances grow and concentrate, so the RBF bandwidth must scale accordingly to remain informative. For two independent samples $\mathbf{x}, \mathbf{y} \sim \mathcal{N}(\mathbf{0}, \mathbf{I}_d)$, for instance, $\mathbb{E}[\|\mathbf{x} - \mathbf{y}\|_2^2] = 2d$ with vanishing relative fluctuations as $d \to \infty$; keeping the RBF exponent $\|\mathbf{x} - \mathbf{y}\|_2^2 / \sigma^2$ of

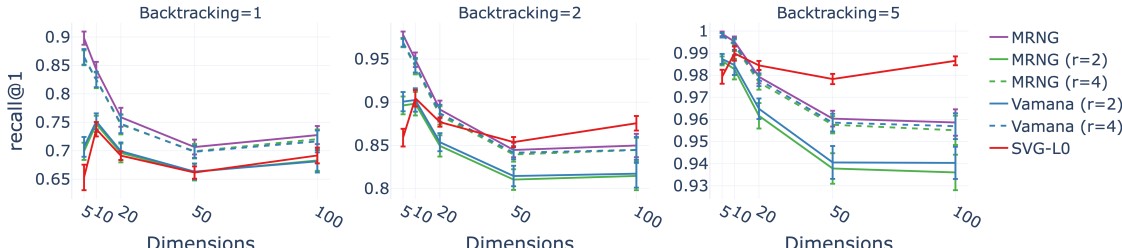

Figure 11: SVG-L0 (Problem (24)) compared against the degree-constrained MRNG (unlimited candidate pool, construction cost quadratic in the number of vectors) and the truncated MRNG and Vamana (candidate-pool ratios $r = 2, 4$), measured by recall@1 over ten realizations of $10^3$ uniformly random vectors for dimensions $d = 5, 10, 20, 50, 100$ using Euclidean-distance similarities. Columns show pure greedy search and backtracking queues of length 2 and 5. The distinguishing behavior is robustness to dimension: once backtracking is used, SVG-L0's recall stays nearly flat as the dimension grows, whereas all baselines, including the degree-constrained MRNG, degrade. Consequently, in higher dimensions SVG-L0 matches or exceeds the degree-constrained MRNG and clearly outperforms the truncated MRNG and Vamana of comparable cost; at backtracking 5 it is the best method at $d = 50$ and $d = 100$. Under pure greedy search it is on par with the truncated baselines and below the degree-constrained MRNG.

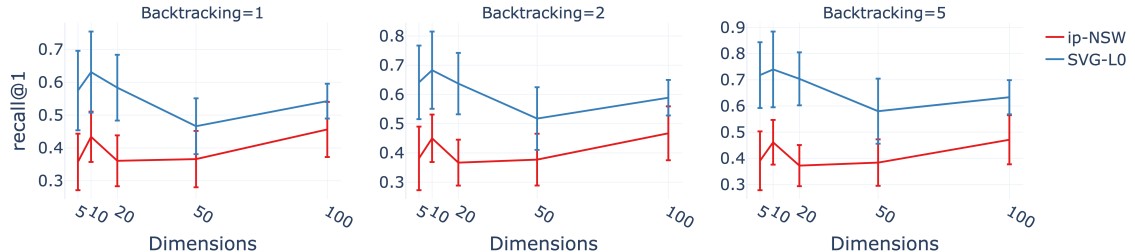

Figure 12: SVG-L0 (Problem (24)) compared against ip-NSW (Morozov & Babenko, 2018), measured by recall@1 over ten realizations of $10^3$ uniformly random vectors for dimensions $d = 5, 10, 20, 50, 100$ using inner-product similarities. Columns show pure greedy search and backtracking queues of length 2 and 5. SVG-L0 outperforms ip-NSW at every dimension and backtracking budget tested, with the margin widening as the backtracking budget grows, consistent with ip-NSW's construction performing no pruning or diversification at all.

order one then requires $\sigma \propto \sqrt{d}$. Hence $\sigma$ need not be tuned blindly: it can be tied to a scale statistic of the data (e.g., the mean or median inter-point distance), which tracks $d$ automatically. This explains why, once $\sigma$ is set to the appropriate scale, SVG-L0's accuracy is largely insensitive to it (Figure 14), including in the high-dimensional regime typical of modern embeddings.

The datasets above are all evaluated under the RBF/exponential kernel ($t = k$). To directly exercise the non-metric branch of our theory on real embeddings, we additionally evaluate SVG-L0 under genuine maximum inner product (MIP) similarity on the Netflix and Yahoo! Music datasets (Table 3), two standard MIPS benchmarks (Morozov & Babenko, 2018). Using the notation $t_k$ of Proposition 1, $t_k \neq k$ for 97.8% of nodes on Netflix and 91.5% on Yahoo! Music, confirming both are genuinely non-metric. We index the 17,770 base vectors for Netflix and a subsample of 20,000 for Yahoo! Music, and evaluate recall on 1,000 held-out queries per dataset (we report 10-recall@10, following the evaluation protocol of Morozov & Babenko (2018)). Figure 15 compares SVG-L0 to ip-NSW (Morozov & Babenko, 2018) at out-degrees $M = 8, 16, 32, 64$. We specifically evaluate the ip-NSW practical top-$M$ construction, not the exact $s$-Delaunay graph their theory covers, which is infeasible to build in high dimensions. SVG-L0 is more accurate than ip-NSW at every out-degree and backtracking budget tested, with the gap starkest at low $M$: on Yahoo! Music at $M = 8$, ip-NSW's recall plateaus near 0.26 regardless of backtracking, while SVG-L0 reaches 0.91, showing that additional backtracking cannot compensate for a poorly connected graph. This improvement stems from

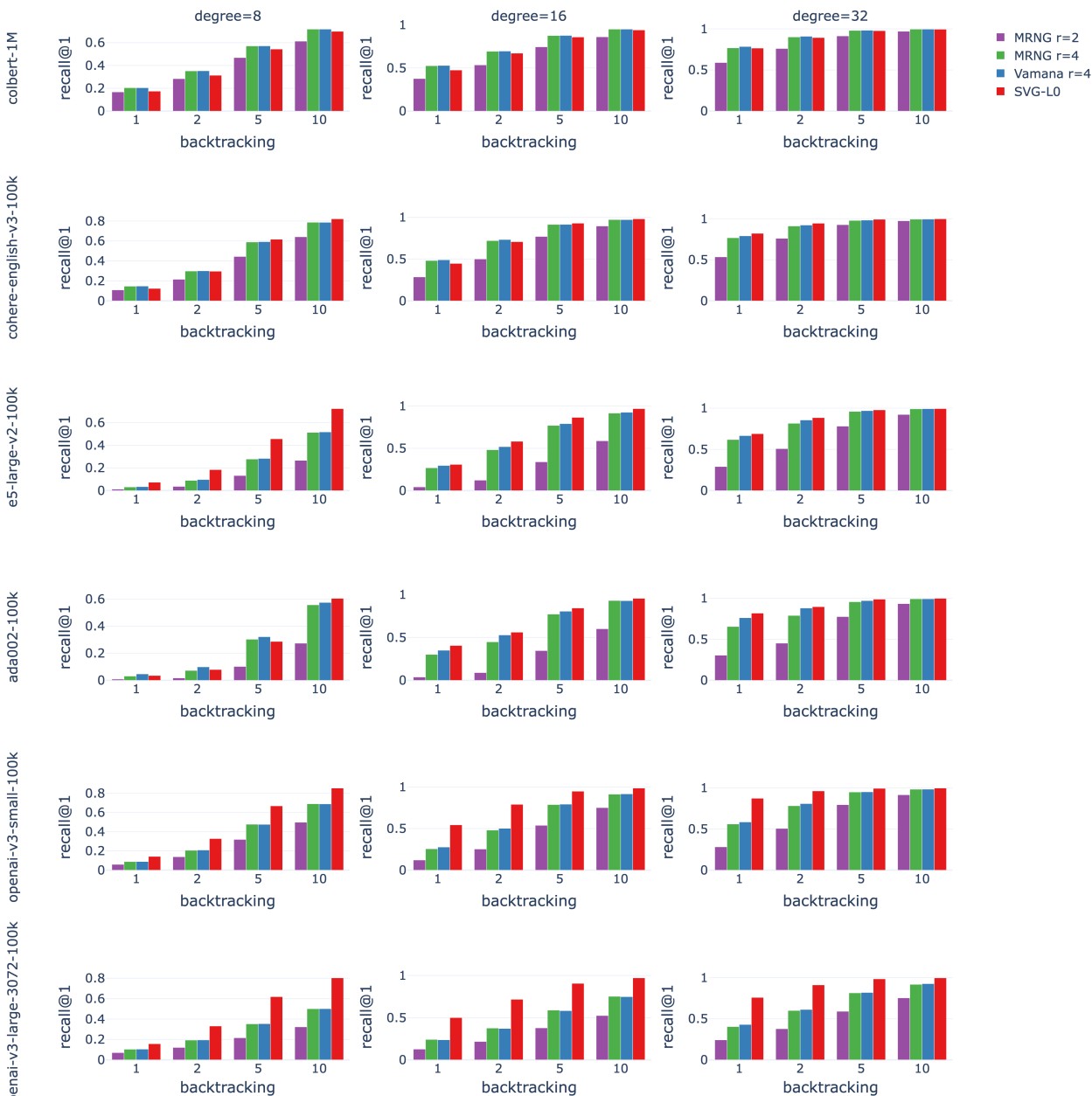

Figure 13: SVG-L0, defined in Problem (24), matches or outperforms the truncated MRNG and Vamana on most datasets and settings, with its advantage growing with the out-degree $M$ and the backtracking budget. The exception is colbert-1M, where SVG-L0 stays on par with the strongest baseline; on ada002-100k it is on par, rather than better, at the lowest out-degree ($M = 8$) and low backtracking, and outperforms the baselines once $M$ or backtracking increases. We compute the recall@1 for different datasets (rows) and maximum out-degrees $M = 8, 16, 32$ (left, center, and right columns, respectively). For the truncated MRNG, we define the truncation ratio $r = |\mathcal{C}|/M$, where $\mathcal{C}$ is the candidate pool. For SVG, we set the number of iterations of Algorithm 4 to $T = 4$ so that it uses the same amount of retrieval as $r = 4$. In practice, finding a suitable $\sigma$ for SVG-L0 is not difficult (automating the selection is left for future work). Full results available in Figure 14 and Appendix D.

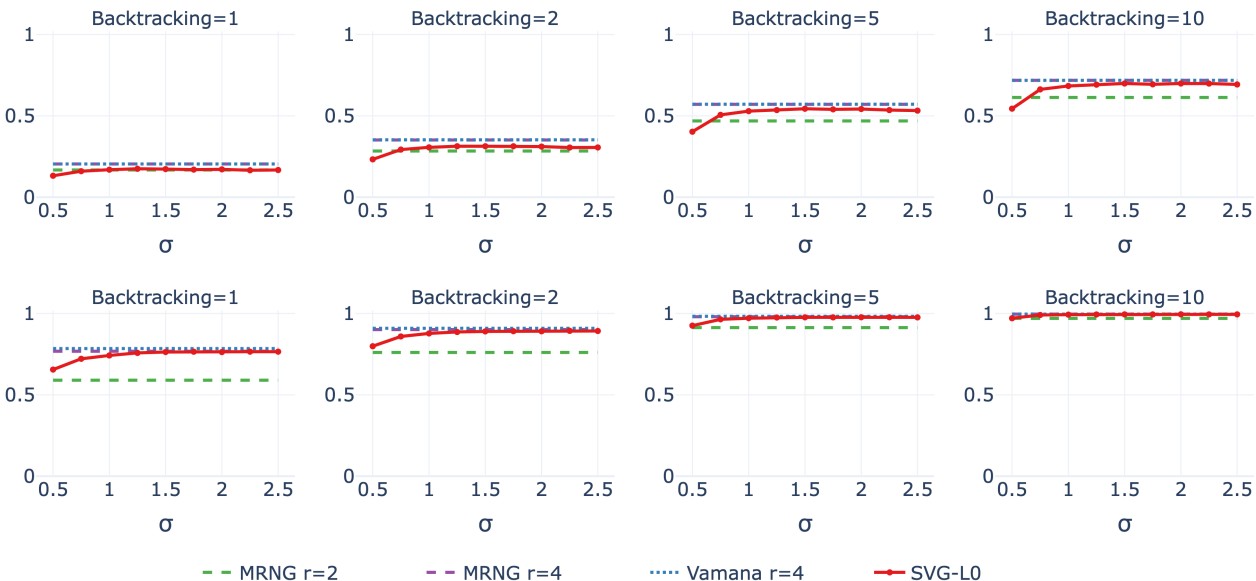

(a) Results on colbert-1M with maximum out-degree $M = 8, 32$ (top and bottom rows, respectively).

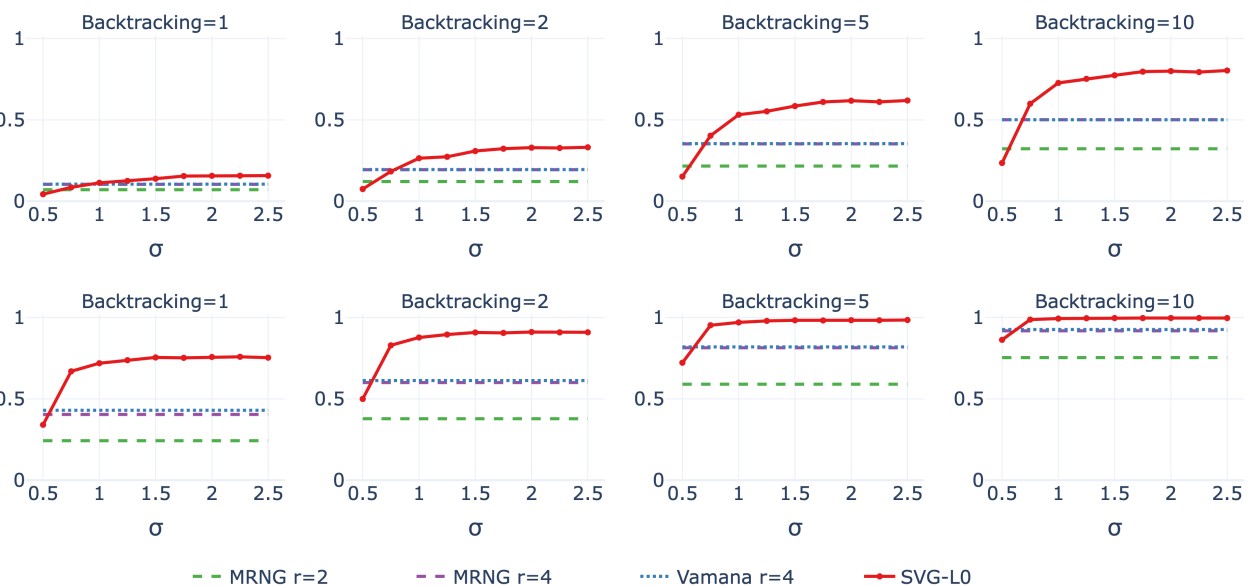

(b) Results on openai-v3-large-3072-100k with maximum out-degree $M = 8, 32$ (top and bottom rows, respectively).

Figure 14: SVG-L0, defined in Problem (24), is shown here on its most and least favorable datasets, across the sweep of the kernel width $\sigma$. On the more favorable dataset (openai-v3-large-3072-100k), it offers better empirical navigability (measured by recall@1) than the truncated MRNG and Vamana at every backtracking budget and out-degree. On the least favorable dataset (colbert-1M), it instead stays on par with the strongest baseline (MRNG $r = 4$ / Vamana $r = 4$), trailing marginally at the lowest out-degree and backtracking and matching it as either increases. For the truncated MRNG, we define the truncation ratio $r = |\mathcal{C}|/M$, where $\mathcal{C}$ is the candidate pool. For SVG, we set the number of iterations of Algorithm 4 to $T = 4$ so that it uses the same amount of retrieval as $r = 4$. In practice, finding a suitable $\sigma$ for SVG-L0 is not difficult (automating the selection is left for future work).

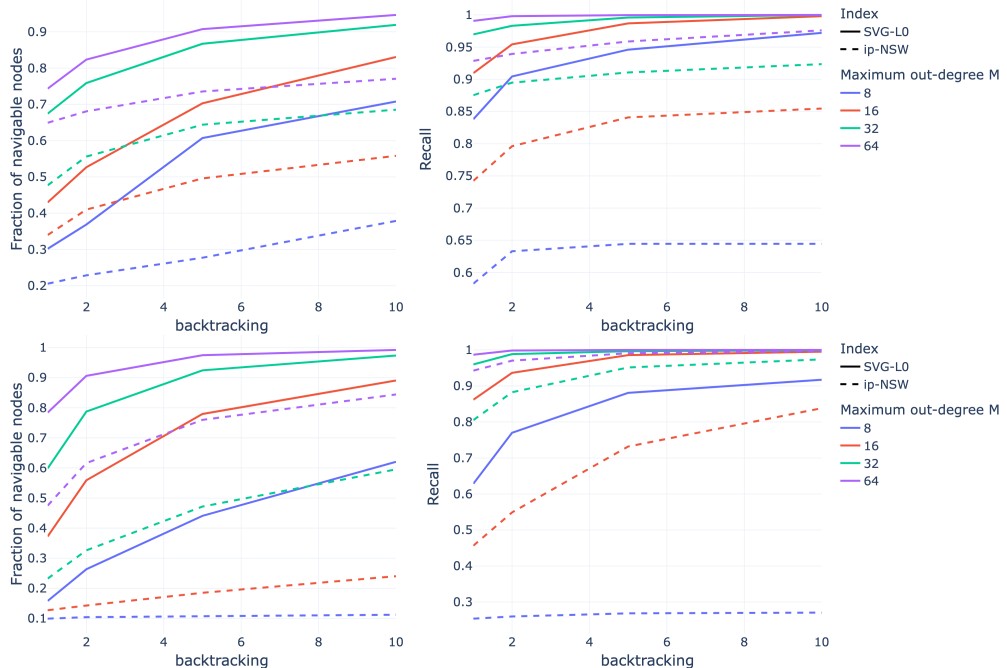

Figure 15: Navigability (left; fraction of nodes $i$ from which a generalized monotonic path reaches $t_i$, Proposition 1) and 10-recall@10 on held-out queries (right) for SVG-L0 and ip-NSW (Morozov & Babenko, 2018) on Netflix (top) and Yahoo! Music (bottom), at out-degrees $M \in \{8, 16, 32, 64\}$ and backtracking budgets $1, 2, 5, 10$. SVG-L0 is more accurate than ip-NSW (the practical top-$M$ construction, not the infeasible exact $s$-Delaunay graph) at every $M$ and backtracking budget tested. The gap is largest at low $M$: on Yahoo! Music at $M = 8$, ip-NSW's recall plateaus near 0.26 regardless of backtracking (additional backtracking cannot compensate for a poorly connected graph), while SVG-L0 reaches 0.91.

SVG-L0's optimization-based edge selection, which favors a diverse set of neighbors; ip-NSW's plain top-$M$ rule has no such mechanism: it performs no pruning or diversification at all.

## 7 Conclusions and future work

We introduced a new type of graph index, the Support Vector Graph (SVG). We derive SVG from a novel perspective that uses machine learning instead of computational geometry to build the index. Concretely, we have formulated the graph construction as a kernelized nonnegative least squares problem. This problem is in turn equivalent to a support vector machine, whose support vectors yield the connectivity of the graph.

We extended the notion of graph navigability to non-Euclidean settings. In this setting, we provide formal navigability results for SVG. Prior work (Morozov & Babenko, 2018) established navigability for the inner-product Delaunay graph, a single non-metric similarity, but that graph is infeasible to build in high dimensions. To our knowledge, SVG is the first concrete, bounded-degree construction that is provably navigable across general metric and non-metric vector spaces.

We formally interpreted the most popular graph indices, including HNSW (Malkov & Yashunin, 2020) and DiskANN (Subramanya et al., 2019), as SVG specializations. We also showed that new traditional (i.e., triangle-pruning) algorithms can be derived from the principles behind this specialization.

Finally, we showed that we can build graphs with a bounded out-degree by adding a sparsity ($\ell_0$) constraint to the SVG optimization, a combination that we name SVG-L0. SVG-L0 yields a principled way of handling the bound, in contrast to the traditional heuristic of simply truncating the out edges of each node. Additionally, SVG-L0 has a self-tuning property, which avoids selecting a candidate set of edges for each graph node and makes its computational complexity sublinear in the number of indexed vectors.

In future work, we plan to address the following issues to further improve SVG and SVG-L0. First, the kernel width does not matter for SVG but it does matter for SVG-L0 empirically. Tuning the width, although not hard and standard in SVMs (e.g., Chapelle et al., 2002), can be challenging in large-scale scenarios. We will address this problem, seeking an automated selection that does not require cross-validation. Second, determining the maximum out-degree for any graph index remains a challenging problem. Interesting insights may be derived from the tools presented in this work. Lastly, it remains to be seen whether other machine learning techniques, beyond kernel methods, can be utilized to build graph indices.

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
