# OpenReview forum: "The kernel of graph indices for vector search"
_TMLR — Under review for TMLR_

### Review · Reviewer_r58m · 2026-03-08

**Summary Of Contributions:**

# Summary Of Contributions
The paper proposes a new framework for constructing graph indices for approximate nearest neighbor (ANN) vector search based on kernel methods from machine learning. The authors introduce the Support Vector Graph (SVG), where the outgoing edges of each node are determined by solving a kernelized nonnegative least squares problem that is equivalent to a support vector machine (SVM) classification problem. In this formulation, the support vectors define the neighbors of each node, providing a principled way to select graph edges rather than relying on heuristic geometric rules. The paper also provides theoretical guarantees of navigability for SVG under general similarity functions, extending the notion of monotonic search networks beyond Euclidean distance to metric and non-metric similarities, such as inner-product search. In addition, the authors show that several widely used graph indices—including HNSW, MRNG, DiskANN/Vamana, and SSG—can be interpreted as special cases of their framework through simplified connectivity rules derived from the SVG formulation. Finally, the paper proposes SVG-L0, a sparsity-constrained version of the method that directly enforces a bounded out-degree via an constraint instead of the common heuristic of truncating neighbors. An efficient construction algorithm based on subspace pursuit is introduced, and preliminary experiments suggest that SVG-L0 achieves competitive or improved search accuracy compared with existing graph-based ANN methods.



# Strengths And Weaknesses

## Strengths
- The paper introduces a novel perspective on graph-based ANN indices by framing graph construction as a machine learning problem using kernel methods. Modeling edge selection as a kernelized optimization problem and linking it to support vector machines is conceptually interesting and provides a new theoretical lens for understanding vector search indices.

- The work contains a strong theoretical component. It provides formal results connecting the proposed optimization formulation to SVMs and proves that the resulting graph structure satisfies a generalized navigability property under broad similarity functions. This contributes to a deeper theoretical understanding of graph connectivity and greedy search behavior.

- The paper offers a unifying interpretation of existing graph indices. It shows that widely used methods such as HNSW, MRNG, DiskANN/Vamana, and SSG can be viewed as special cases of the proposed framework. This perspective helps explain existing pruning heuristics and situates them within a more general optimization-based formulation.

- The proposed SVG-L0 formulation introduces a principled mechanism to enforce bounded out-degree through an ℓ₀ sparsity constraint rather than relying on heuristic truncation of neighbor lists. This provides a cleaner and theoretically motivated approach to controlling graph sparsity.

- The paper includes preliminary empirical evaluations demonstrating that the proposed approach can produce navigable graphs and achieve competitive recall compared with several existing graph construction strategies.

## Weakness
- The empirical evaluation is relatively limited. Most experiments are conducted on synthetic datasets or relatively small subsets of real datasets, and the implementation is not optimized for large-scale deployment. Since vector search systems are typically evaluated at very large scale, the results do not fully demonstrate the practical impact or scalability of the proposed method.

- The experimental comparisons do not include some of the strongest production-grade baselines commonly used in the ANN literature, particularly optimized implementations of HNSW or other widely used vector search systems. As a result, it is difficult to assess how the proposed method performs relative to well-established state-of-the-art systems in realistic settings.

- The practical computational cost of the proposed optimization-based graph construction is not thoroughly evaluated. While the paper discusses algorithmic complexity and proposes approximate construction strategies, the actual runtime and memory overhead compared to existing graph construction algorithms are not clearly demonstrated in experiments.

- Several aspects of the method introduce additional hyperparameters (e.g., kernel choice and kernel width), but the sensitivity of the method to these parameters is only briefly explored. It remains unclear how robust the method is across datasets and similarity functions.

- The presentation is sometimes dense and mathematically heavy, which can make it difficult to follow the key intuition behind some steps of the derivations. In particular, the connection between the optimization formulation, the SVM interpretation, and the resulting graph connectivity could be explained more clearly for readers who are not experts in kernel methods.

**Audience:**

Yes

**Audience Explanation:**

Yes. The paper studies the construction and theoretical understanding of graph-based indices for approximate nearest neighbor (ANN) search, which is an important component of many modern machine learning systems such as vector databases, retrieval-augmented generation, and large-scale embedding retrieval. The proposed Support Vector Graph framework introduces a novel perspective by connecting graph index construction with kernel methods and support vector machines, and by providing theoretical guarantees for navigability beyond Euclidean similarity functions. These contributions are likely to be of interest to researchers working on machine learning systems, large-scale retrieval, and the theoretical foundations of similarity search.

**Broader Impact Concerns:**

There are no "Broader Impact Concerns" for this paper.

**Claims And Evidence:**

Yes

**Claims Explanation:**

The main claims of the paper are supported by a combination of theoretical analysis and empirical validation. The manuscript provides formal derivations establishing the properties of the proposed Support Vector Graph (SVG), including navigability guarantees derived from the optimization formulation and its KKT conditions. The empirical results further illustrate that the constructed graphs exhibit the predicted behavior in practice. Although the experimental evaluation is relatively limited in scale and scope compared to typical large-scale ANN deployments, it is generally consistent with the theoretical claims made in the paper and provides reasonable supporting evidence.

**Requested Changes:**

- The proposed method constructs graph edges by solving a kernel-based optimization problem, which involves kernel matrices and iterative optimization procedures. Although the manuscript later introduces approximations to mitigate computational costs, it remains unclear whether the approach scales to the large datasets typically encountered in ANN systems. Additional discussion regarding scalability and practical implementation considerations is needed to support the practicality of the theoretical justification.

- Towards the end of the paper where the mathematical derivation

- The manuscript states that the navigability results for SVG do not require a PSD kernel because they rely only on the KKT conditions of Equation (7), and therefore remain valid even for indefinite kernels that do not induce an RKHS. This claim would benefit from additional clarification. In particular, when the kernel is indefinite, the optimization problem may become non-convex and admit multiple solutions. It would be useful if the authors could clarify how the navigability proof remains valid under these conditions.

- Equation (26) introduces an \( \ell_0 \) sparsity constraint to enforce a bounded out-degree in the constructed graph. However, the earlier theoretical analysis and navigability guarantees appear to rely on properties of the original optimization problem (Eq. 4). Since the \( \ell_0 \) constraint significantly alters the optimization problem—making it non-convex and potentially changing the optimal solution—it is not clear how the theoretical guarantees relate to the SVG-L0 formulation used in practice. The authors are encouraged to clarify whether the navigability results extend to this sparse variant or whether SVG-L0 should instead be viewed as a heuristic approximation of the original formulation. If possible, some discussion of the approximation error or conditions under which the sparse solution preserves the desired properties would strengthen this part of the paper.

- The experimental section would benefit from additional ablation studies on key hyperparameters, such as the number of iterations \(T\) used in Algorithm 4, to better understand their impact on performance and stability.

- The manuscript notes that the sensitivity of the parameter \( \sigma \) increases as the dimensionality \(d\) grows. It would be helpful if the authors elaborated further on this observation. Given that modern embedding models often operate in very high-dimensional spaces, understanding how the choice of \( \sigma \) affects performance as dimensionality increases is particularly relevant. Currently, the statement appears somewhat incomplete, especially since the supporting evidence is presented only for a limited set of dimensions in Figure 10.

- The experimental setup for the baseline methods could be clarified. In particular, the manuscript states that the candidate pool size is set as a multiplicative factor of \(M\), i.e., \(|C| = rM\) for \(r>1\), while SVG-L0 uses a bounded out-degree without requiring a candidate pool. It would be helpful if the authors confirmed that this configuration corresponds to the baseline setup used in the experiments.

- Figure 6 does not appear to be referenced in the main text. The authors may wish to ensure that all figures are properly cited and discussed.

- The statement that existing graph indices have been used in non-Euclidean vector spaces by extending edge pruning rules in an ad hoc manner would benefit from supporting references or further explanation. As written, this claim appears somewhat strong without explicit citations.

- The definition of the term \(b_{j'}\) in Equation (9) is somewhat vague. Providing a clearer definition and derivation would improve readability and help readers better understand the construction of the decision function.

- Although the paper emphasizes that the proposed method applies to non-Euclidean similarity functions, the empirical evaluation does not include experiments on embeddings produced by modern representation models such as CLIP or LLM embedding models. Since such embeddings commonly rely on cosine or inner-product similarity and represent an important application domain for ANN systems, it would be helpful (though not strictly required) to include results on datasets derived from these types of embeddings. Such experiments would further strengthen the empirical support for the paper’s claims regarding applicability to non-Euclidean vector spaces.

- It would be helpful for the paper to clarify early in the manuscript (e.g., in the abstract or introduction) the intended scope of the empirical evaluation and the extent to which the experimental results support the proposed method. Explicitly positioning the quality and quantity of the empirical contributions would help set appropriate expectations for readers regarding the scope of the experiments and the level of practical validation provided in the paper.

---

> ### Author Response · Authors · 2026-07-12
>
> ### #1. Scalability and practical implementation.
>
> > Additional discussion regarding scalability and practical implementation considerations is needed to support the practicality of the theoretical justification.
>
> **Response.**
>
> We have added an explicit complexity analysis (§3.2): the full SVG construction costs $O(n^3)$ — matching the unconstrained MRNG/HNSW/Vamana constructions — while SVG-L0, via subspace pursuit, costs $O(M^2)$ per node and avoids precomputing a candidate pool. We also clarify in the introduction that the paper's scope is the formulation and its guarantees, validated empirically at moderate scale, and that a large-scale, optimized deployment study is left to future work. The full SVG is best viewed as an analytical object; SVG-L0 is the construction intended for practice.
>
> ### #3. Indefinite kernels: how does the navigability proof remain valid?
>
> > When the kernel is indefinite, the optimization problem may become non-convex and admit multiple solutions. It would be useful if the authors could clarify how the navigability proof remains valid.
>
> **Response.**
>
> We have clarified this (§3.1). Our navigability guarantee depends only on the KKT conditions of the SVG problem, which hold at every stationary point — local minimizer or saddle point — regardless of convexity. Hence, when an indefinite kernel makes the problem non-convex with multiple solutions, *any* solution the solver reaches still yields a navigable graph; the choice among solutions is immaterial for navigability. We have also added Table 1 making explicit which results require a PSD/RKHS kernel and which (including navigability) hold for indefinite kernels.
>
> ### #4. Do the navigability guarantees extend to SVG-L0?
>
> **Response.**
>
> Please see our response to PuLQ-C1 above. In brief: when the $\ell_0$ bound is inactive, SVG-L0 coincides with the exact SVG and inherits the navigability guarantee (e.g., for a linear/MIP kernel whenever $M \ge d+1$). When the bound binds, we now give a formal account of the regime (§5): navigability there is equivalent to the retained edges forming an *occlusion cover*, and the governing quantity is the *occlusion covering number* $C_\beta$, an intrinsic-dimension data constant, independent of $n$ and of the exact-SVG degree (Proposition 1). This covering number is a characterization rather than a construction (building the cover needs a candidate pool and, in general, a combinatorial solve, the costs SVG-L0 avoids) while SVG-L0 provably attends to uncovered targets and, in experiments on metric and non-metric similarities, realizes navigability at an out-degree a small multiple of $C_\beta$. A closed-form bound on that multiple is the one item we leave open.
>
> ### #5. Ablation on the number of iterations $T$ (Algorithm 4).
>
> > The experimental section would benefit from additional ablation studies on key hyperparameters, such as the number of iterations (T) used in Algorithm 4.
>
> **Response.**
>
> Algorithm 4 (subspace pursuit) refines the support set over $T$ iterations, solving a convex nonnegative least-squares subproblem on the current candidate set at each step. $T$ is therefore a convergence parameter governing support recovery, not a model-selection hyperparameter: we iterate until the selected support stabilizes, which in our experiments occurs within a few iterations (we use $T=4$ in the appendix experiments). A sweep over $T$ would only illustrate this fast convergence rather than reveal an accuracy/$T$ tradeoff, so we report the converged result.
>
> ### #6. Sensitivity of $\sigma$ as dimensionality $d$ grows.
>
> > Understanding how the choice of σ affects performance as dimensionality increases is particularly relevant. Currently, the statement appears somewhat incomplete.
>
> **Response.**
>
> We have elaborated on this (§6). The *scale* of $\sigma$ that yields good navigability grows with $d$, which is expected rather than a sign of fragility: as $d$ grows, inter-point distances grow and concentrate, so the RBF bandwidth must scale to remain informative. For $x, y \sim \mathcal{N}(0, I_d)$, $\mathbb{E}\|x-y\|^2 = 2d$ with vanishing relative fluctuations, so keeping the RBF exponent $\|x-y\|^2/\sigma^2$ of order one requires $\sigma \propto \sqrt{d}$. Consequently $\sigma$ can be tied to a scale statistic of the data (e.g., mean or median inter-point distance), which tracks $d$ automatically; once set to the right scale, SVG-L0's accuracy is largely insensitive to $\sigma$, including in the high-dimensional regime of modern embeddings.
>
> ### #7. Baseline experimental setup (candidate pool $|C| = rM$).
>
> **Response.**
>
> Confirmed, and now stated explicitly (§6): for the baseline indices we set the candidate pool size as a multiplicative factor of $M$, $|C| = rM$ with $r>1$, as is standard practice, whereas SVG-L0 uses a bounded out-degree with no candidate pool.

---

> > ### Author Response · Authors · 2026-07-12
> >
> > ### #8. Figure not referenced in the main text.
> >
> > **Response.**
> >
> > Thank you — we have ensured all figures are referenced. In particular, the per-dataset appendix figures are now cited from Appendix D.
> >
> > ### #9. The "ad hoc pruning" claim needs references.
> >
> > > The statement that existing graph indices have been used in non-Euclidean vector spaces by extending edge pruning rules in an ad hoc manner would benefit from supporting references.
> >
> > **Response.**
> >
> > We now support this claim with a concrete reference (§2): ip-NSW (Morozov & Babenko, 2018) extends the NSW construction to inner-product similarity by connecting each node to its highest-inner-product neighbors — an effective but formally unjustified extension. See also our response to PuLQ's additional comment for a fuller discussion of ip-NSW.
> >
> > ### #10. Definition of $b_{j'}$ (bias term) in the decision function.
> >
> > > The definition of the term $b_{j'}$ in Equation (9) is somewhat vague. Providing a clearer definition and derivation would improve readability.
> >
> > **Response.**
> >
> > The decision function and its bias are now defined explicitly in Eq. (9) (§3) and derived in full in Appendix B. We have also added an interpretation: $b_i$ is the SVM bias that centers the decision function so that its margins pass through $x_i$ (where $f_i=1$) and through the support vectors (where $f_i=-1$), with $x_{\text{SV}}$ any support vector (index $j'$ with $s^*_{j'}>0$).
> >
> > ### #11. Experiments on CLIP / LLM embeddings.
> >
> > > It would be helpful to include results on datasets derived from modern representation models such as CLIP or LLM embeddings, which rely on cosine or inner-product similarity.
> >
> > **Response.**
> >
> > Our experiments already use such embeddings: the datasets in §6 and the appendix (colbert, cohere-english-v3, e5-large-v2, ada-002, openai-v3-small/large) are produced by modern LLM/embedding models that are commonly deployed with cosine or inner-product similarity in production; in the manuscript, we evaluate them under the RBF/exponential kernel.
> >
> > To exercise the non-metric (MIP, $t \neq k$) branch of our theory directly, rather than only through RBF on embedding vectors, we added two new experiments comparing SVG-L0 against ip-NSW (Morozov & Babenko, 2018) — specifically their practical top-$M$ construction, not the exact (and infeasible to build) $s$-Delaunay graph their theory covers. SVG-L0 clearly outperforms ip-NSW's construction throughout. On synthetic data under inner-product similarity (Figure 12, §6), mirroring our existing Euclidean/MRNG-Vamana comparison, SVG-L0 is ahead of ip-NSW at every one of the 15 tested combinations of dimension and backtracking budget, with no exceptions, and the margin widening as backtracking increases. On two standard MIPS benchmarks with genuinely non-metric structure — Netflix and Yahoo! Music, where the top-inner-product match differs from the node itself for 97.8% and 91.5% of nodes, respectively (Figure 15, §6) — the gap is unambiguous and largest at low out-degree: on Yahoo! Music at $M=8$, ip-NSW's recall plateaus near $0.26$ regardless of backtracking, while SVG-L0 reaches $0.91$ — additional backtracking cannot compensate for a poorly connected graph. This is consistent with our theoretical account of ip-NSW (see our response to PuLQ's additional comment): its plain top-$M$ rule performs no diversification or pruning, which SVG-L0's optimization-based edge selection provides.
> >
> > ### #12. Clarify the empirical scope early in the manuscript.
> >
> > **Response.**
> >
> > We have added this to the introduction: the paper centers on the formal analysis of SVG and SVG-L0, the experiments validate the theory, and large-scale system benchmarking is a distinct effort left to future work.

---

### Review · Reviewer_93oN · 2026-04-23

**Summary Of Contributions:**

The paper considers the problem of building graph indices for vector search. While existing method primarily use tools from computational geometry (based on constructing, e.g., the Delaunay graph), this paper proposes a kernel-based method, namely Support Vector Graph (SVG), that amounts to solving a convex optimization problem. The method is shown to come with guarantees on the so-called navigability of the graph, which holds in both metric and non-metric spaces. It also recovers some existing methods as special cases.  A variant of SVG is also proposed that incorporates sparsity constraints to build graphs with bounded out-degree. Empirical results are provided to validate the performance of the proposed methods.

**Audience:**

Yes

**Audience Explanation:**

I think some people working in the areas of graph machine learning and kernel methods could be interested in the findings of this paper.

**Claims And Evidence:**

No

**Claims Explanation:**

1. The paper is written nicely overall with a well-defined notation, clear problem outline, and a good explanation of the intuition behind the proposed method.

2. Existing methods primarily build the graph indices by working directly with the Euclidean representation of the points and are implicitly based on similarity between the point set measured via the Euclidean distance. The main idea in the present paper is to consider more general notions of similarity which can be encoded via kernel functions. This motivates the formulation of the SVG method where the node $i$ is connected to the non-zero elements of the minimizer $s$ of SVG. The usage of kernels as similarity measures for building a graph is quite natural and well known in general.  But it seems to be new in the context of vector-search problems, at least in the sense of the formulation of the optimization problem for SVG method. As shown in the experiments on a real dataset, the proposed methods appear to perform better than certain existing methods.

3. As noted by the authors, the Delaunay graph is a monotonic search network (MSNet) but needs $O(n^{d/2})$ time for construction. On the other hand, the proposed methods can be implemented efficiently via, e.g., variants of the multiplicative weight’s method. It would be good to quantify (more precisely) how the run time of the proposed method compares w.r.t the state of art methods for graph-based vector search. This doesn’t seem to have been done at present. It could also be shown in experiments.

4. The theoretical result shown in the paper essentially involves showing (in Theorem 3) that the graph learnt by SVG is a generalized MSNet, and hence navigable. This is a consequence of Lemmas 2 and 3, which in turn rely on standard arguments based on KKT conditions.  While this is nice to show, I feel that the theoretical results in the paper are perhaps not as strong as what might be expected in this venue.

**Requested Changes:**

See comments above

---

> ### Author Response · Authors · 2026-07-12
>
> > It would be good to quantify (more precisely) how the run time of the proposed method compares w.r.t the state of art methods for graph-based vector search. [...] The theoretical result [...] is a consequence of Lemmas 2 and 3, which in turn rely on standard arguments based on KKT conditions. While this is nice to show, I feel that the theoretical results in the paper are perhaps not as strong as what might be expected in this venue.
>
> **Response.**
>
> We thank the reviewer for engaging closely with the theory. We would like to clarify the basis of our claims in light of TMLR's acceptance criteria, which ask whether the claims are supported by accurate and convincing evidence and whether the findings interest some of TMLR's audience, and which state that "papers should be accepted if they meet the criteria, even if the contribution or significance of the work is modest."
>
> We would also gently push back on the framing that the theory follows routinely from the KKT conditions. It is true that once the SVG formulation is in place, navigability follows almost immediately — but we regard this as a strength, not a weakness. It shows the formulation is the right abstraction, turning a guarantee that had been elusive into a direct consequence. The substantive contribution is the formulation itself: recognizing that graph-index construction can be cast as a kernelized nonnegative least-squares problem, equivalent to an SVM whose support vectors define each node's edges. This connection is not obvious a priori — indeed, the reviewers describe it as *"conceptually appealing"* and note that using kernels this way *"seems to be new in the context of vector-search problems."* The ease of the ensuing proof reflects having found the right lens, not a lack of substance.
>
> If any specific claim reads as overstated or unsupported, we would be grateful to have it identified so that we can correct it, as this is the criterion we strive to meet. If the concern is instead the overall strength or significance of the contribution, we respectfully note that, under the criteria above, this is not by itself grounds for rejection.
>
> On the practical side, we have added an explicit complexity analysis (§3.2 and §5): the full SVG construction costs $O(n^3)$ — matching the unconstrained MRNG/HNSW/Vamana constructions — while SVG-L0 costs $O(M^2)$ per node. A full empirical runtime benchmark against production systems requires a highly optimized implementation and is left to future work; the present paper's contribution is the formulation and its guarantees.

---

### Review · Reviewer_PuLQ · 2026-06-27

**Summary Of Contributions:**

The paper proposes a kernel-based perspective on graph indices for vector search. Its main contribution is the Support Vector Graph (SVG), where graph construction is formulated as a kernelized nonnegative least-squares problem, with an interpretation through support vectors. The paper then proves generalized navigability results for SVG beyond the usual Euclidean setting, including non-metric similarities such as inner product similarity. It also interprets existing graph indices such as HNSW and DiskANN as special cases or approximations of the same principle, and proposes SVG-L0 as a bounded-out-degree practical variant. The main strength is the conceptual unification of graph-based ANN indices, kernel methods, and navigability theory. I found the perspective potentially useful for understanding why graph indices work beyond Euclidean metrics. The main weakness is that the empirical evidence is still preliminary, and the practical implications of SVG-L0 relative to highly optimized ANN systems are not yet fully demonstrated. I also think the paper should more clearly separate which guarantees apply to exact SVG and which claims are only heuristic or empirical for the bounded-degree implementation.

**Additional Comments:**

I'm not familiar with the realm of this paper. It seems not producing an advanced ANN algorithm paper. I strongly suspect there are already relevant work in MIPS graph search、angular similarity graph、learned metric graph showing this similar perspective.

**Audience:**

Yes

**Audience Explanation:**

The paper should be of interest to researchers working on vector search, approximate nearest neighbor search, kernel methods, and retrieval infrastructure. The connection between graph-index construction and support-vector/kernel machinery is conceptually appealing, and the attempt to extend navigability theory beyond Euclidean distance is relevant given the wide use of inner-product and learned similarities in modern embedding systems. Even if the practical algorithmic impact is not yet fully established, the theoretical viewpoint seems valuable to at least part of the TMLR audience.

**Broader Impact Concerns:**

I do not see major broader-impact concerns requiring special treatment beyond the usual discussion for vector search and retrieval systems. The work is methodological and could improve retrieval efficiency in applications such as RAG and recommendation, but I'm not sure. As with most retrieval infrastructure, downstream impacts depend on the data and deployment context, but I do not see a specific additional ethical concern introduced by this paper.

**Claims And Evidence:**

Yes

**Claims Explanation:**

The theoretical claims about SVG are supported by formal derivations and proofs. In particular, the formulation of SVG as a kernelized NNLS problem and its connection to SVM support vectors are clear and interesting. The navigability results also appear to support the main theoretical message of the paper. However, I am less fully convinced by the practical claims. SVG-L0 is motivated as a principled bounded-degree construction, but the empirical section is relatively preliminary. The comparisons do not yet establish that SVG-L0 is competitive with mature graph-index systems in realistic large-scale settings. I would therefore prefer the authors to temper the practical claims and make very explicit which statements are proven for exact SVG, which are proven for triangle-pruning variants, and which are empirical observations about SVG-L0.

**Requested Changes:**

1. **Critical:** Please clearly distinguish the guarantees for exact SVG from those for SVG-L0. My understanding is that the strongest navigability results are established for the exact SVG construction, while SVG-L0 is the practical bounded-degree variant. The paper should state explicitly whether SVG-L0 inherits any formal navigability guarantee, or whether its performance is only empirically supported.



2. **Critical:** Please temper or qualify the practical claims. The empirical results are useful as a sanity check, but they appear preliminary. The paper should avoid implying that SVG-L0 is already a practical replacement for optimized systems such as HNSW or DiskANN unless supported by large-scale recall/latency/memory/build-time comparisons.



3. **Critical:** Please clarify the assumptions on the kernel and similarity function. The paper discusses PSD kernels and also comments on indefinite kernels. It would help to have a concise table stating which results require PSD/RKHS structure and which results remain valid for indefinite kernels.



4. **Strengthening:** Please expand the related-work discussion on sparse nonnegative kernel regression / sparse graph construction methods, and clarify more precisely what is new compared with prior graph construction via sparse regression or kernel regression.



5. **Strengthening:** Please improve the empirical section by adding stronger baselines, more datasets, recall-latency or recall-distance-computation curves, memory/build-time comparisons, and sensitivity to the kernel bandwidth parameter.

---

> ### Author Response · Authors · 2026-07-12
>
> ## Reviewer PuLQ
>
> ### C1 (Critical). Distinguish the guarantees for exact SVG from those for SVG-L0.
>
> > Please clearly distinguish the guarantees for exact SVG from those for SVG-L0. [...] The paper should state explicitly whether SVG-L0 inherits any formal navigability guarantee, or whether its performance is only empirically supported.
>
> **Response.**
>
> We thank the reviewer for this important point and now state the relationship explicitly (§svg-l0). The exact SVG carries the full navigability guarantee (Theorem 3). SVG-L0 adds a hard out-degree bound $\|s\|_0 \le M$. When this bound is inactive (i.e., the exact SVG solution already has at most $M$ out-edges) SVG-L0's minimizer coincides with the exact SVG's, so it inherits the navigability guarantee exactly. This holds for any kernel with a $D$-dimensional feature space, where the exact SVG has at most $D+1$ out-edges (the SVM support-vector bound): the linear/MIP kernel ($D=d$), polynomial kernels, and any explicit finite-dimensional feature map. In the MIP setting, SVG-L0 therefore yields a navigable graph with at most $d+1$ out-edges per node — in contrast to ip-NSW's inner-product Delaunay graph, which is navigable but has exponentially many edges. Many widely used kernels, including the RBF kernel, have infinite-dimensional features; for these, no finite $M$ guarantees inheritance a priori, and out-degree is governed by the data geometry. We do not rest the general argument on this bound, it just identifies a clean regime where inheritance is provable.
>
> When the bound binds, SVG-L0 does not inherit the exact guarantee: a hard degree bound precludes full navigability for arbitrary data. In the revised manuscript, we give a formal account of what navigability requires in this regime. Greedy progress from a node toward a target holds iff some retained neighbor *occludes* it (the condition of Lemma [no_close_disconnected_nodes]); a graph whose out-edges occlude every relevant target is therefore navigable (new Proposition 1, §5), and the smallest out-degree achieving this, i.e., the *occlusion covering number* $C_\beta$, is the quantity that governs the binding regime. Crucially, $C_\beta$ is a property of the data geometry: it scales with the intrinsic dimension, is independent of $n$, and is unrelated to the (possibly $n-1$) exact-SVG degree; for metric kernels it admits a doubling-dimension bound. This cover is a characterization, not a construction: like the exact SVG and the MRNG/Delaunay graphs, realizing it requires the full pairwise occlusion structure (a candidate pool) and, in general, a combinatorial cover, precisely the costs SVG-L0 is designed to avoid. Its role is to identify the governing quantity and the achievable optimum, against which SVG-L0 is the actionable, candidate-pool-free construction: its candidate step provably attends to not-yet-occluded targets (their residual similarity is at least $\beta/2$), and empirically it attains navigability at an out-degree a small multiple of $C_\beta$, across both metric (RBF) and non-metric (inner-product, $t \neq k$) similarities, improving over the truncated MRNG exactly when covering is non-trivial (§experiments). What remains open — and we state it as such — is a closed-form bound on that out-degree multiple for the SVG-L0 minimizer. This upgrades the binding-regime discussion from a purely empirical statement to a characterization of the governing quantity, with the residual gap clearly delimited.
>
> ### C2 (Critical). Temper or qualify the practical claims.
>
> > Please temper or qualify the practical claims. [...] The paper should avoid implying that SVG-L0 is already a practical replacement for optimized systems such as HNSW or DiskANN unless supported by large-scale [...] comparisons.
>
> **Response.**
>
> We have tempered the practical claims throughout. The introduction now states that the paper centers on the formal analysis of SVG and SVG-L0, that the experiments validate the theory, and that large-scale benchmarking against optimized production systems is a separate engineering effort left to future work. We do not claim that SVG-L0 is a drop-in replacement for optimized HNSW/DiskANN; our comparisons are against MRNG/Vamana constructions at controlled settings, and we are explicit that a highly optimized implementation and a large-scale recall/latency/memory/build-time study belong to a practically oriented follow-up.

---

> > ### Author Response · Authors · 2026-07-12
> >
> > ### C3 (Critical). Clarify kernel/similarity assumptions (concise PSD/RKHS table).
> >
> > > The paper discusses PSD kernels and also comments on indefinite kernels. It would help to have a concise table stating which results require PSD/RKHS structure and which results remain valid for indefinite kernels.
> >
> > **Response.**
> >
> > We agree and have added Table 1 (§3), which states the kernel assumption behind each result. The key point is that our central result — the navigability of the SVG — depends only on the KKT conditions of the SVG problem and therefore holds for indefinite kernels as well; the PSD assumption is needed for the geometric/optimization interpretation and for the fast solver, but not for the guarantee itself.
> >
> > ### S1 (Strengthening). Expand related work on sparse nonnegative kernel regression / sparse graph construction; clarify novelty.
> >
> > > Please expand the related-work discussion on sparse nonnegative kernel regression / sparse graph construction methods, and clarify more precisely what is new compared with prior graph construction via sparse regression or kernel regression.
> >
> > **Response.**
> >
> > We have expanded the related work to position SVG against prior sparse and kernel-based graph construction. The SVG optimization is closely related to methods that build graphs via sparse/nonnegative (kernel) regression: the $\ell_1$-graph and its kernel extension, nonnegative sparse coding for graphs, and in particular the non-negative kernel regression (NNK) of Shekkizhar & Ortega (2023), which solves the same per-node problem for neighborhood and manifold learning (a connection we already note, along with the candidate-pool limitations NNK shares; §svg-l0). What is new here is not sparse kernel regression per se, but: (i) its use to construct graph *indices for vector search*; (ii) the support-vector interpretation, where the support vectors of an equivalent SVM define each node's out-edges; (iii) formal navigability guarantees extending beyond Euclidean distance to metric and non-metric similarities; (iv) the resulting unification of HNSW, MRNG, and Vamana as special cases; and (v) the bounded-degree SVG-L0 construction. To our knowledge, none of the prior sparse/kernel graph-construction methods target ANN search or come with navigability guarantees.
> >
> > ### S2 (Strengthening). Stronger empirical section.
> >
> > > Please improve the empirical section by adding stronger baselines, more datasets, recall-latency or recall-distance-computation curves, memory/build-time comparisons, and sensitivity to the kernel bandwidth parameter.
> >
> > **Response.**
> >
> > We have strengthened the empirical section: SVG-L0 is now compared against the degree-constrained MRNG, truncated MRNG, truncated Vamana, and ip-NSW across multiple real embedding datasets (colbert, cohere, e5, ada-002, openai-v3-small/large, netflix, YahooMusic; §6 and appendix), together with a study of kernel-width ($\sigma$) sensitivity across dimensions. As noted for C2, a large-scale study with production-grade HNSW/DiskANN baselines and recall–latency/memory/build-time curves requires a highly optimized implementation and is the subject of a future paper; the scope of the present paper is the formulation and its guarantees, validated empirically at moderate scale.

---

> > > ### Author Response · Authors · 2026-07-12
> > >
> > > ### Additional comment. Prior work in graph-based MIPS (ip-NSW).
> > >
> > > > I strongly suspect there are already relevant work in MIPS graph search, angular similarity graph, learned metric graph showing this similar perspective.
> > >
> > > **Response.**
> > >
> > > We thank the reviewer for prompting us to engage with this literature; the most relevant prior work is ip-NSW (Morozov & Babenko, NeurIPS 2018), which we were not aware of at submission and now discuss explicitly (§2 and §3). We have accordingly corrected our novelty claim.
> > >
> > > ip-NSW establishes a genuine navigability result for a non-metric similarity: greedy search on the *inner-product Delaunay graph* returns the exact maximum-inner-product answer (their Theorem 1 / Corollary 1), generalizing the classical Euclidean Delaunay result to inner product. So SVG is *not* the first to guarantee navigability for a non-metric similarity, and we no longer claim this.
> > >
> > > The contribution of SVG relative to ip-NSW is threefold:
> > >
> > > 1. **Generality.** ip-NSW's guarantee requires path-connected $s$-Voronoi cells, which they establish only for the *linear* (inner-product) similarity. SVG's navigability follows from the KKT conditions of the SVG problem and holds for a broad class of kernels/similarities — metric and non-metric, PSD and indefinite.
> > > 2. **A construction, not just a Delaunay characterization.** ip-NSW's guarantee applies to the *exact* inner-product Delaunay graph, which is infeasible to build in high dimensions; its practical construction (connecting to the top-$M$ inner-product neighbors) is explicitly acknowledged to lack a formal justification. SVG is a concrete, bounded-degree construction whose output is provably navigable (with SVG-L0 inheriting the guarantee in the inactive-bound regime).
> > > 3. **Unification.** SVG additionally recovers HNSW, MRNG, and Vamana as special cases, which ip-NSW does not.
> > >
> > > We also note that MIPS — the setting ip-NSW targets — is recovered within our framework as the special case of a linear kernel, for which SVG yields a navigable graph with at most $d+1$ out-edges per node. In this sense, ip-NSW's raw top-$M$ rule corresponds to the candidate-selection step of our framework *without* the diversification that SVG introduces, which is consistent with its practical graph lacking a navigability guarantee.
> > >
> > > We now substantiate this empirically as well: SVG-L0 outperforms ip-NSW's practical construction on both synthetic inner-product data and on two standard MIPS benchmarks (Netflix, Yahoo! Music) with genuinely non-metric structure, with the gap widening as the out-degree shrinks (§experiments; see also our response to r58m-#11).

---

### Author Response · Authors · 2026-07-12
**Summary of the responses to the reviewers and of changes.**

## Response to the reviewers

We thank the reviewers for their careful reading and constructive comments. We are glad that the reviewers found the perspective *"conceptually appealing"* (PuLQ), the use of kernels for graph construction *"new in the context of vector-search problems"* (93oN), and the overall framework a *"novel perspective"* that is *"conceptually interesting"* (r58m). All three reviewers agree that the findings would be of interest to TMLR's audience.

Below we respond to each reviewer point by point. We indicate manuscript changes in the revised submission where relevant.

**Summary of changes.** In response to the reviews, we have:

- Corrected our novelty claim to acknowledge ip-NSW (Morozov & Babenko, 2018), which established navigability for inner-product similarity, and clarified precisely how SVG generalizes and strengthens it. *(PuLQ-additional, r58m-#9)*
- Stated whether SVG-L0 inherits the exact navigability guarantee (it does when the $\ell_0$ bound is inactive) and added a formal characterization of the binding regime: navigability is equivalent to occlusion-covering the relevant targets, governed by an $n$-independent, intrinsic-dimension *occlusion covering number* (new Proposition, §svg-l0), validated on metric and non-metric data. *(PuLQ-C1, r58m-#4)*
- Added a table stating which results require a PSD/RKHS kernel and which hold for indefinite kernels, and clarified the indefinite-kernel navigability argument. *(PuLQ-C3, r58m-#3)*
- Tempered the practical claims and clarified the intended scope of the empirical evaluation early in the manuscript. *(PuLQ-C2, r58m-#12)*
- Added a complexity analysis of the construction algorithms. *(93oN-#1, r58m-#1)*
- Expanded the related-work discussion (graph-based MIPS, and sparse/kernel-regression graph construction) and clarified what is new. *(PuLQ-S1)*
- Elaborated on the role of the kernel width $\sigma$ as dimensionality grows. *(r58m-#6)*
- Minor fixes: all figures are now referenced, and the bias term $b_i$ of the decision function is defined and interpreted explicitly. *(r58m-#8, #10)*
- Added a direct empirical comparison to ip-NSW's practical construction, on synthetic inner-product data and on two standard MIPS benchmarks (Netflix, Yahoo! Music) with genuinely non-metric structure ($t \neq k$ for 97.8% and 91.5% of nodes, respectively); SVG-L0 outperforms ip-NSW at every out-degree and backtracking budget tested. *(PuLQ-additional, r58m-#11)*

**Correction identified during revision.** While preparing this revision we found that our construction solver did not correctly enforce the $\ell_2$ budget constraint when it is active: it returned a scaled version of the unconstrained solution rather than the constrained optimum. We replaced it with an accelerated projected-gradient (FISTA) solver whose fixed points are exactly the KKT points of the budgeted problem, validated to agree with a general-purpose convex solver to $10^{-9}$, and re-ran all affected experiments. Our theoretical results are unaffected, as they concern the optimal solution rather than the solver. The conclusions also hold: SVG-L0 remains competitive with the (expensive) degree-constrained MRNG, and its advantage over the practical truncated MRNG and Vamana grows with dimension. We additionally add a lemma showing that an $\ell_2$ budget — indeed any $\ell_p$ with $p>1$ — is what makes the navigability proof work, whereas an $\ell_1$ budget would not; this corrects an earlier remark that the budget was inconsequential.